# On the Detection of Reviewer-Author Collusion Rings From Paper Bidding

**Steven Jecmen**                                                            *sjecmen@gmail.com*
*Carnegie Mellon University*

**Nihar B. Shah**                                                            *nihars@cs.cmu.edu*
*Carnegie Mellon University*

**Fei Fang**                                                                 *feifang@cmu.edu*
*Carnegie Mellon University*

**Leman Akoglu**                                                             *lakoglu@andrew.cmu.edu*
*Carnegie Mellon University*

**Reviewed on OpenReview:** *https://openreview.net/forum?id=o58uy91V2V*

## Abstract

Collusion rings pose a significant threat to peer review. In these rings, reviewers who are also authors coordinate to manipulate paper assignments, often by strategically bidding on each other's papers. A promising solution is to detect collusion through these manipulated bids, enabling conferences to take appropriate action. However, while methods exist for detecting other types of fraud, no research has yet shown that identifying collusion rings is feasible.

In this work, we consider the question of whether it is feasible to detect collusion rings from the paper bidding. We conduct an empirical analysis of two realistic conference bidding datasets and evaluate existing algorithms for fraud detection in other applications. We find that collusion rings can achieve considerable success at manipulating the paper assignment while remaining hidden from detection: for example, in one dataset, undetected colluders are able to achieve assignment to up to 30% of the papers authored by other colluders. In addition, when 10 colluders bid on all of each other's papers, no detection algorithm outputs a group of reviewers with more than 31% overlap with the true colluders. These results suggest that collusion cannot be effectively detected from the bidding using popular existing tools, demonstrating the need to develop more complex detection algorithms as well as those that leverage additional metadata (e.g., reviewer-paper text-similarity scores).

## 1 Introduction

In scientific peer review, researchers are called upon to evaluate the work of other researchers in the same scientific community and provide a recommendation as to whether each work should be published. Within the field of computer science, academic conferences are some of the primary venues of publication. These conferences receive a large pool of paper submissions from authors, and they rely on a peer review process to accurately assess the quality of each submission and determine which to accept or reject. The resulting acceptance decisions can have substantial impacts on the careers of the authors, as publication of a paper in a competitive computer science conference is often used as an indication of research quality. Thus, it is vital that the peer review processes of these conferences are fair to the authors.

However, the perceived prestige of a conference publication can incentivize authors and reviewers to act in unethical ways. One such form of unethical behavior that has occurred in computer science conferences is

that of *collusion rings* (Vijaykumar, 2020; Littman, 2021). Conferences recruit many (if not all) qualified authors of submitted papers to serve as reviewers, and consequently many reviewers at a conference are also authors of one or more submissions to that same conference. In a collusion ring, a group of reviewers and authors at a particular conference coordinate to manipulate the paper assignment such that they are assigned to review each other's papers. The most straightforward way that colluding reviewers can manipulate the assignment is through "paper bidding," a common phase of the paper assignment process. During the paper bidding phase, each reviewer is asked to express their level of interest in reviewing each paper. These bids are then taken into account when determining the paper assignment and often have a large influence (Jecmen et al., 2020; Wu et al., 2021; Leyton-Brown et al., 2022). Thus, colluders can and do use strategic bidding to achieve a desired paper assignment (Littman, 2021):

> *"The colluders hide conflicts of interest, then bid to review these papers, sometimes from duplicate accounts, in an attempt to be assigned to these papers as reviewers."*

Once assigned, the colluders can give positive reviews to and push for the acceptance of other colluders' papers.

Approaches to addressing academic fraud can be divided into mitigation-based and detection-based methods (Wilson, 2020). Several past works have proposed modifications to paper assignment algorithms that aim at mitigating the impact of malicious bidding (Guo et al., 2018; Jecmen et al., 2020; Wu et al., 2021; Boehmer et al., 2022; Jecmen et al., 2022). However, such approaches necessarily come with a cost in terms of assignment quality, as they ignore some aspects of reviewer preferences expressed through bids. Moreover, they do not identify which (if any) individuals were engaged in collusion. In contrast, detection methods have received little attention in prior work. Ideally, an accurate detection method could identify any colluders so that the conference can intervene to stop them. For example, a conference could decide to block the assignment of detected colluders to each other's papers, potentially leading to higher-quality assignments among the honest reviewers as compared to mitigation methods. Reliable detection methods could also allow the conference organizers to take more serious action against the colluding individuals, perhaps after a manual investigation provides further evidence. In the words of Littman (2021):

> *"Better paper-assignment technology would help close one loophole that is being exploited. But, without better investigative tools, we may never be able to hold the colluders to account."*

However, careless deployment of detection algorithms may result in false positives: honest reviewers and authors being falsely identified as colluders. This is a serious danger in scientific peer review, since falsely-accused reviewers and authors may have their reputations tarnished or their papers unfairly rejected. Establishing effective methods to detect collusion rings is thus a critical path for research on this problem.

A long line of research has applied anomaly-detection techniques to successfully detect various forms of fraud in other settings. Many of these settings involve a network of people in which the fraud appears as a set of anomalous interactions: for example, auction fraud on online platforms, fraudulent financial transactions, fake reviews on sites such as Amazon, and fake accounts on social media (Akoglu et al., 2015). This raises an important question of whether these techniques can be similarly applied to effectively detect fraudulent interactions in the context of paper bidding for peer review. However, there is a vast range of approaches to detection in the research literature that could potentially be applied to this problem. To facilitate a thorough analysis, our work focuses on methods that attempt to detect collusion using only the paper bidding and authorship data (i.e., the data that directly reflects the strategic bidding of colluders). Our focus on bidding is additionally motivated by the fact that real-world investigations of collusion often begin with analysis of the bidding data. Furthermore, many conferences use bidding-based heuristics to mitigate collusion (e.g., ignoring the bids of reviewers with too few positive bids; Jecmen et al. 2022). Concretely, we consider the question: **is it possible to effectively detect collusion rings from only the bids and authorships?** Answering this question alone requires an extensive empirical analysis. However, resolving this question provides important direction for future research on detecting collusion rings. Our work does not attempt to investigate various other questions of interest to the problem of collusion detection: for example, comparing the effectiveness of detection- and mitigation-based approaches, conducting a game-theoretical analysis of

the detection problem, or gathering new data on the behavior of real colluders. As additional motivation for our focus on using only bidding data, many conferences in practice use only bids to determine the paper assignment (with the potential addition of some coarse subject-area matching); for instance, this is the case for most conferences hosted on the popular HotCRP conference management system (where the first instances of collusion were caught).

Within the realm of detection algorithms, we consider algorithms that attempt to accurately identify a single group of colluders rather than those that identify a list of several potentially-colluding groups with lower confidence. While an algorithm outputting multiple groups would be more likely to output the colluding group, the aforementioned cost of false positives in our setting means that conferences should rightly be hesitant to accuse a reviewer of collusion without a high degree of certainty. Detection algorithms with a high false-positive rate could instead be applied as part of a collusion-mitigation approach in which a large number of reviewer-paper pairs are flagged as potentially collusive and blocked from being assigned. However, we view this class of algorithms as falling within the "mitigation" category rather than the "detection" category we focus on, and as such should be compared to to the other collusion-mitigation approaches that have been proposed in the literature.

One major challenge in the peer review setting is that there is no ground-truth data about how colluding reviewers behave in practice. In this work, instead of assuming a specific form of collusive bidding behavior, we address this challenge by considering a wide range of parameterized behavior for the collusion rings. Specifically, since the purpose of a collusion ring is to achieve colluder-to-colluder paper assignments, we vary the density of bidding between colluding reviewers and colluder-authored papers. Denser bidding corresponds to a stronger attack, but also may allow the colluding group to be more easily detected. Thus, our analysis aims to establish the regions of this parameterized space of colluder behavior at which collusion can be effectively detected.

Our contribution is a set of empirical results on two different realistic bidding datasets concerning the feasibility of detecting collusion rings. Our analysis consists of three parts. **First**, we characterize the typical density of bidding found within groups of honest (non-colluding) reviewers, as high-density groups of honest reviewers are potential false positives for detection algorithms. **Second**, we evaluate the performance of several established algorithms at detecting injected collusion rings based on the anomalous density of bidding within the ring. We consider a combination of fundamental approaches to finding dense subgraphs and density-based fraud detection methods, including two algorithms that have been shown to effectively detect fraud in other settings. **Third**, we evaluate the success of the injected collusion rings at achieving their desired paper assignments, contextualizing the potential impact of fraud.

Overall, our results suggest that collusion rings **cannot** be effectively detected from only the bidding and authorship data using popular existing algorithms. Our findings include the following:

- Using only the bidding and authorship data, all detection algorithms we analyze fail to detect some injected collusion rings that are larger than any honest-reviewer groups with a similar density of bidding. For example, when the collusion ring consists of 10 reviewers who bid on all of each other's authored papers, the output of the best-performing detection algorithm across both datasets has only a 31% overlap (Jaccard similarity) with the true colluders on average.

- Colluders are able to achieve assignment to a substantial fraction of the papers authored by other colluders while avoiding detection by all analyzed algorithms (30% and 24% in each of the two datasets).

- A sizeable fraction of colluders can get at least one of their papers reviewed by another colluder while avoiding detection (54% and 35% in each of the two datasets).

These results suggest that future research on detecting collusion rings must focus on more complex detection methods, especially those that leverage other metadata (such as reviewer-paper text-similarity scores or reviewer publication history).

The code and data we use in our analysis are available online at `https://github.com/sjecmen/peer-review-collusion-detection`.

## 2 Related Work

Recent research has looked at improving various aspects of the peer review process (Shah, 2022). In this line of work, various solutions have been proposed to the problem of reviewer-author collusion that we consider (Jecmen et al., 2022). Jecmen et al. (2020) propose an algorithm for randomizing the paper assignment such that the probability of a colluder achieving assignment to another colluder's paper is limited. Wu et al. (2021) propose learning a model of reviewer preferences from the bidding and using this model to determine the paper assignment, thereby reducing the influence of collusive bidding. This work also provides a semi-synthetic paper bidding dataset, which we use in our analysis. The works (Guo et al., 2018; Boehmer et al., 2022) consider the problem of finding a cycle-free paper assignment. In the 2021 AAAI Conference on Artificial Intelligence, a large AI conference, several precautions were taken to address the problem of collusion: two-cycles in the paper assignment were disallowed and a constraint on the countries of the reviewers assigned to the same paper was added (Leyton-Brown et al., 2022). Notably, these solutions all aim to mitigate the impact of collusion rings on the assignment, rather than detecting them outright. While the present work focuses on detecting attacks in the paper bidding phase, other works show that attacks on the text-similarity algorithms used by conferences can be effective (Markwood et al., 2017; Tran & Jaiswal, 2019; Eisenhofer et al., 2023).

Most relevant to our work is the prior work (Jecmen et al., 2023), which observes paper bidding in a mock conference setting and collects data on the strategies employed by participants acting as colluding reviewers. In contrast, rather than attempting to precisely simulate the behavior of colluding reviewers, we consider an intentionally simplified setting where malicious reviewer behavior can be easily parameterized, allowing us to sweep out a wide range of potential colluder strategies. We note that the most common colluder strategies observed by Jecmen et al. (2023) consist of injecting dense bidding between colluders, as is done by the strategies considered in our analysis. Jecmen et al. (2023) also evaluate the performance of very simple detection methods on the collected dataset, while we consider a set of more complex detection methods based on dense-subgraph discovery.

Outside of the setting of scientific peer review, similar problems of fraud have been studied in the anomaly detection literature. In online platforms such as Amazon or Yelp, several methods have been proposed to detect products or sellers who purchase fraudulent reviews from users (Akoglu et al., 2013; Eswaran et al., 2017; Kumar et al., 2018). Other works propose density-based methods for detecting groups of fraudsters in these and similar online network settings (e.g., fraudulent transactions on eBay or fake followers on Twitter) (Pandit et al., 2007; Prakash et al., 2010; Hooi et al., 2016); we evaluate the performance of (Hooi et al., 2016) in our analysis. However, our setting (scientific peer review) is distinct from these online platform settings in a few ways. Most significantly, our work focuses on collusion in the paper bidding phase, where the objective of colluders is to achieve a desired outcome in the subsequent paper assignment; in contrast, the fraudulent reviews are themselves the objective of fraudsters on (e.g.) Amazon. As a result, the incentives for peer-review colluders are not the same as those of fraudsters in the other settings: for example, since each reviewer is assigned to review a limited number of papers, "camouflaging" by bidding positively on non-colluder papers may result in those papers being assigned instead of the targeted papers. In addition, fraudulent interactions on online platforms are often provided by large numbers of fake accounts specifically used for fraud; since making fake accounts on common peer-review platforms is non-trivial, interactions between colluders in peer review may be more often done under their real identities, leading to different patterns of behavior. Numerous works have proposed other methods for graph-based anomaly detection, including those that address online spam, cyber-attacks, and other forms of fraud (Akoglu et al., 2015). Generic approaches for dense-subgraph detection (Goldberg, 1984; Tsourakakis et al., 2013; Hooi et al., 2020) can also be applied to detect anomalies in graphs; we evaluate these methods in our analysis.

Aside from reviewer-author collusion rings, reviewers can attempt to dishonestly improve the chances of acceptance of their own papers in other ways. One such form of malicious behavior occurs when reviewers provide negative reviews to the papers they are assigned, thereby improving the relative standing of their own authored work. In contrast to the issue of collusion rings, which require the cooperation of multiple reviewers, this sort of "strategic reviewing" can be done by each reviewer alone. In a controlled experiment, Balietti et al. (2016) found that people do exhibit strategic reviewing in a competitive peer evaluation setting.

Several works have proposed methods to mitigate strategic reviewing with theoretical guarantees (Alon et al., 2011; Xu et al., 2019; Dhull et al., 2022). On the detection side, Stelmakh et al. (2021) design a statistical test for the presence of strategic reviewing and evaluate it on data collected from a peer evaluation experiment.

## 3 Preliminaries

In this section, we detail the setting of our analysis, the datasets we analyze, and the research questions we answer.

### 3.1 Setting

We consider a conference peer review setting with a set of submitted papers $\mathcal{P}$ and a set of reviewers $\mathcal{R}$. Conferences generally recruit the authors of the submitted papers to serve as reviewers, along with external, non-author reviewers. The authorship set $\mathcal{A} \subset \mathcal{R} \times \mathcal{P}$ contains all pairs $(r, p)$ such that reviewer $r$ authored paper $p$. Additionally, the conflict-of-interest set $\mathcal{C} \subset \mathcal{R} \times \mathcal{P}$ contains all pairs $(r, p)$ such that reviewer $r$ has a conflict-of-interest with paper $p$ and should not be assigned to review it ($\mathcal{A} \subseteq \mathcal{C}$). Once submissions are received, the conference asks each reviewer to indicate their level of interest on each submitted paper via paper bidding. While conferences often allow each reviewer to choose from a number of discrete levels (e.g., "Eager", "Willing", "Not willing"), we consider a simplified setting with binary bids (positive or neutral); note that allowing additional levels of bids can only give colluders more flexibility to manipulate the paper assignment. The bid set $\mathcal{B} \subset \mathcal{R} \times \mathcal{P}$ contains all pairs $(r, p)$ such that reviewer $r$ bid positively on paper $p$ ($\mathcal{B} \cap \mathcal{C} = \emptyset$). The conference also computes text-similarity scores between each reviewer and paper using a function $T : \mathcal{R} \times \mathcal{P} \to [0, 1]$, where $T(r, p)$ indicates the level of similarity between the text of paper $p$ and the text of the past work of reviewer $r$.

The conference then computes the paper assignment in the following manner. First, the conference computes similarity scores between each reviewer and paper using a function $S : \mathcal{R} \times \mathcal{P} \to [0, 1]$. In our experiments, we use $S(r, p) = \frac{1}{2}T(r, p)2^{\mathbb{I}[(r,p)\in\mathcal{B}]}$ based on the function used in the 2016 Conference on Neural Information Processing Systems (NeurIPS), a large machine-learning conference (Shah et al., 2018). The conference then computes an assignment of papers to reviewers such that the total similarity of the assigned pairs is maximized, subject to constraints that (i) each paper is assigned to exactly 3 reviewers, (ii) each reviewer is assigned at most 6 papers, and (iii) no reviewer-paper pairs in $\mathcal{C}$ are assigned. While we use the stated values in our experiments, the exact reviewer and paper loads vary between conferences. The maximum-similarity assignment can be computed efficiently as a min-cost flow or as a linear program. This framework for paper assignment has been used in numerous conferences and venues (Shah, 2022).

### 3.2 Datasets

We next provide details regarding the two datasets that we analyze in this work.

The first dataset, which we refer to as "AAMAS", contains a subset of the real bidding from the 2021 International Conference on Autonomous Agents and Multiagent Systems, an AI conference. This dataset is publicly available from PrefLib (Mattei & Walsh, 2013) and contains de-identified bids from reviewers that did not opt-out from data collection. In this conference, bids were selected from {"Yes", "Maybe", "Conflict", "No response"}; we consider "Yes" and "Maybe" responses to be positive bids and include those reviewer-paper pairs in $\mathcal{B}$. We set $\mathcal{C}$ as the set of all reviewer-paper pairs with "Conflict" bids. Since the dataset does not include authorship information, we reconstruct the authorships $\mathcal{A}$ by subsampling 3 conflicts-of-interest uniformly at random for each paper. The resulting dataset has 526 papers and 596 reviewers, 398 of whom authored at least one paper. The dataset also does not contain text-similarity scores $T(r, p)$. We generate synthetic text-similarity scores using the procedure described in Appendix A, based on the text-similarities from the 2018 International Conference on Learning Representations reconstructed by Xu et al. (2019).

Our second dataset, which we refer to as "S2ORC", is the semi-synthetic dataset constructed and made publicly available by Wu et al. (2021). This dataset contains synthetic bids between a large subset of published computer science papers and authors from the Semantic Scholar Open Research Corpus (Ammar

et al., 2018), designed to match statistics from the NeurIPS 2016 conference (Shah et al., 2018). Bids were chosen from values $\{0, 1, 2, 3\}$, where non-zero values indicate a positive bid (included in $\mathcal{B}$). For the authorship set $\mathcal{A}$, we use the real authorships between the reviewers and papers in the dataset, discarding the 90 bids placed on self-authored papers. We assume that the conflicts-of-interest are only the authorships ($\mathcal{C} = \mathcal{A}$). The resulting data has 2446 papers and 2483 reviewers, 984 of whom authored at least one paper. This dataset also contains real text-similarity scores between each reviewer and paper $T(r, p)$, computed using the popular TPMS algorithm (Charlin & Zemel, 2013). We compare the level of agreement between these text-similarity scores and the bids to those found by Stelmakh et al. (2023) and find that they have a realistic level of error; see Appendix A for details.

In our analysis, we assume that both of these datasets contain only bids from "honest" (non-colluding) reviewers. The AAMAS dataset contains information from reviewers that did not opt-out from the data collection, and we expect that any colluding reviewers in the conference would have done so. The S2ORC bids are synthetic and do not model malicious reviewer behavior.

### 3.3 Problem Statement

Our goal is to detect collusion rings. We suppose that there exists a group of colluding reviewers $\mathcal{M} \subset \mathcal{R}$ who try to manipulate the paper assignment by altering their bids, with the aim of being assigned to review the papers authored by other members of the colluding group. The objective of a collusion-detection algorithm is to output $\mathcal{M}$ given the set of bids $\mathcal{B}$, along with the authorships $\mathcal{A}$ and the other conflicts $\mathcal{C}$. Note that the text-similarities $T(r, p)$ cannot be used for detection in our analysis–we consider the problem of detection using only the bidding itself and the authorships.

As our original datasets do not contain colluding reviewers, we inject collusion into the datasets by choosing a group of reviewers to be the colluders $\mathcal{M}$ and modifying the bids of these reviewers (i.e., adding or removing elements of $\mathcal{M} \times \mathcal{P}$ to/from $\mathcal{B}$). The strongest form of collusion would be to add bids between all reviewers in $\mathcal{M}$ and all papers authored by other reviewers in $\mathcal{M}$, while additionally removing all other bids by reviewers in $\mathcal{M}$. However, malicious reviewers may not want to perform such an obvious manipulation out of fear of being caught. Due to the lack of concrete evidence on colluding groups and the exact bidding strategy that they might employ, we consider a wide range of possible colluding groups parameterized by both a size parameter (i.e., the number of colluding reviewers) and a "density" parameter (roughly corresponding to the attack strength).

As there are some modeling choices involved in concretely defining the bidding density parameter, we consider two different notions of density, corresponding to two different graph representations of the bidding relationships between the reviewers of the conference. The first representation (Section 4) is a unipartite graph in which vertices represent reviewers. The second representation (Section 5) is a bipartite graph in which vertices represent both reviewers and papers. We motivate and define each of these formulations in their respective sections. For each graph formulation, we investigate the following three high-level research questions:

- **Q1:** For what values of size and density do groups of honest reviewers already exist in the dataset (without injected collusion)? (Sections 4.1 and 5.1)

- **Q2:** For what values of size and density can groups of injected colluders be accurately detected by existing algorithms? (Sections 4.2 and 5.2)

- **Q3:** At each size and density, how successful are groups of injected colluders at achieving their desired paper assignments? (Sections 4.3 and 5.3)

All of our experiments in these sections were conducted on a server with 515 GB RAM and 112 CPUs.

## 4 Unipartite Bidding Graph

In this section, we represent the reviewer bidding data in the form of a unipartite, directed graph where each vertex corresponds to a reviewer. The graph contains a directed edge $(r_1, r_2)$ if reviewer $r_1$ bid on at

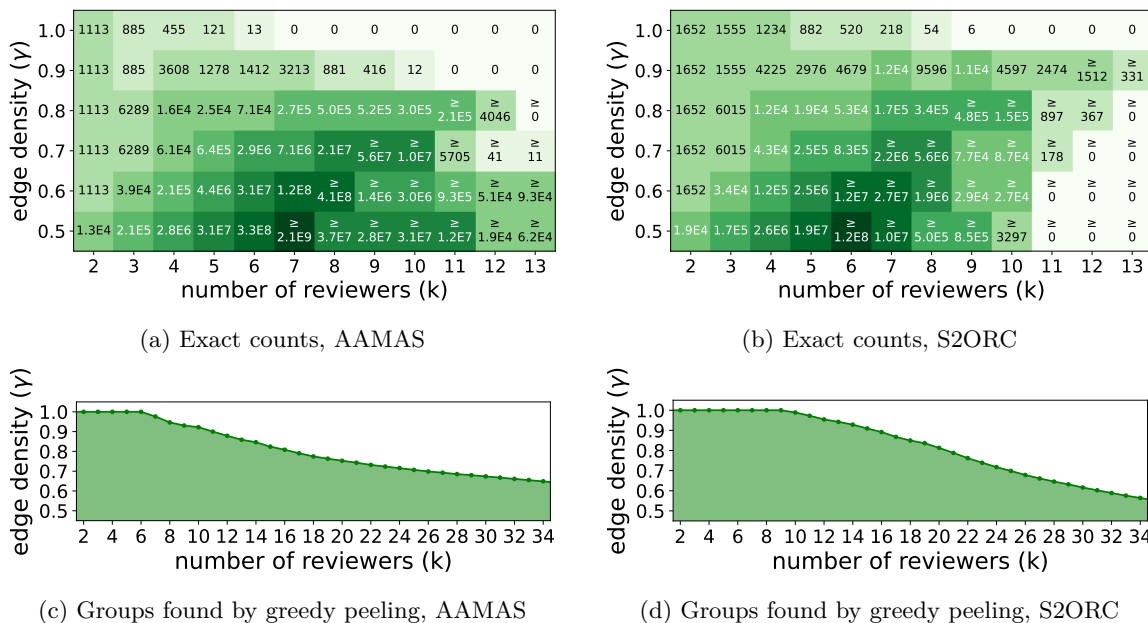

Figure 1: Exact counts of honest-reviewer groups with varying size and edge density (Figures 1a-1b), and the size and edge density of honest-reviewer groups found by a heuristic method (Figures 1c-1d). In Figures 1a-1b, values in cells marked with "≥" represent lower bounds since exact counts were infeasible to compute; the vertical axis represents lower bounds on the edge density. In Figures 1c-1d, each point corresponds to an existing honest-reviewer group (found by a greedy peeling method), and the shaded area indicates the region in which there exists at least one honest-reviewer group. The vertical axis represents exact values for the edge density. Note that the vertical axis does not start at 0 for easier comparison with other figures.

least one paper authored by reviewer $r_2$. Informally, the density of a group of reviewers in this graph is the fraction of possible edges present between the reviewers. We formalize these definitions shortly. In Section 5, we consider an alternative, bipartite graph representation of the bidding.

Our motivation for considering this graph representation is that it most naturally represents the reciprocal behavior expected of a collusion ring. Several existing works (Guo et al., 2018; Boehmer et al., 2022; Leyton-Brown et al., 2022) on reviewer-author collusion consider a similar reviewer-to-reviewer graph, aiming to prevent cycles (i.e., "rings") between reviewers in the paper assignment. For our purposes, the notion of density implied by this graph has several useful properties. First, malicious reviewers may not attempt to manipulate the assignment of every paper they author. For example, each colluder might have one bad paper that they want their fellow colluders to review (e.g., because it is lower quality) and many good papers not involved in the manipulation. Even if each colluder authors many good papers, collusion would still appear as a dense subgraph. Additionally, collusion rings are based on a quid-pro-quo arrangement between the colluders, where each colluder needs help getting some papers accepted to the conference. Each colluder therefore should receive some benefit from the other colluders, reflected as density.

Formally, we denote the (directed) graph by $\mathcal{G}_1 = (\mathcal{V}_1, \mathcal{E}_1)$, where $\mathcal{V}_1 = \mathcal{R}$ and $\mathcal{E}_1 = \cup_{p \in \mathcal{P}} \{(r_1, r_2) \in \mathcal{R}^2 : (r_1, p) \in \mathcal{B} \wedge (r_2, p) \in \mathcal{A}\}$. Given this graph representation, we characterize a potential group of colluding reviewers $\mathcal{S} \subseteq \mathcal{R}$ in terms of two parameters. The first parameter $k_{\mathcal{S}} = |\mathcal{S}|$ is the size of the group. The second parameter $\gamma_{\mathcal{S}}$ is the "edge density" of the group, defined as follows. For an arbitrary graph $\mathcal{G} = (\mathcal{V}, \mathcal{E})$ and any subset of vertices $\mathcal{V}' \subseteq \mathcal{V}$, we use $\mathcal{E}[\mathcal{V}']$ to denote the set of edges in the subgraph induced by $\mathcal{V}'$. We then define the edge density of $\mathcal{S}$ to be $\gamma_{\mathcal{S}} = \frac{|\mathcal{E}[\mathcal{S}]|}{2\binom{|\mathcal{S}|}{2}}$, the fraction of possible edges present. We henceforth omit the subscript $\mathcal{S}$ from $k_{\mathcal{S}}$ and $\gamma_{\mathcal{S}}$ since the subset in question will be clear from context.

We now analyze the problem of detecting collusion from $\mathcal{G}_1$. In the following three subsections, we provide empirical analysis to answer each of the three research questions identified in Section 3.3.

### 4.1 Honest-Reviewer Groups (Q1)

In this subsection, we characterize the density of bidding among groups of honest (i.e., non-colluding) reviewers in our datasets. For each value of the parameters $(k, \gamma)$, we count the number of groups of size $k$ with edge density at least $\gamma$. Since we aim to detect collusion among authors, we consider only reviewers that have authored at least one paper. This shows the frontier of identifiable detection: if honest-reviewer groups exist at some $(k, \gamma)$, then a colluding group with that same size and density cannot be identified as suspicious with high confidence based on the bidding within the colluding group. Stated differently, this analysis gives the number of potential false positives for a detection algorithm at each $(k, \gamma)$. Recall that false positives are a serious concern in collusion detection due to the danger to honest reviewers' reputations.

In Figures 1a-1b, we show the results of this analysis. These counts were obtained via a standard backtracking search. Cells marked with "$\geq$" were unable to be completed in 24 hours, as obtaining exact counts takes worst-case time exponential in $k$. Instead, the values in these cells represent lower-bounds. We note that these figures may be independently useful to conferences, since many conferences use heuristic checks for collusion based on reviewer bidding patterns as part of the paper assignment (e.g., not assigning certain reviewer-paper pairs or ignoring certain reviewers' bids). The results in Figures 1a-1b may help these conferences understand the false positive rates of such heuristics.

Since exact counts are infeasible for large values of $k$, we additionally use a heuristic method to find dense subgraphs of larger sizes. We start with the entire graph and iteratively remove the vertex of smallest degree to produce a sequence of subsets of decreasing size (commonly called "greedy peeling"). In Figures 1c-1d, we plot the edge density $\gamma$ of these subsets against the size $k$. Thus, for all points $(k, \gamma)$ in the shaded region, at least one group of honest reviewers exists of size $k$ and with edge density at least $\gamma$.

Overall, these results show that there is a large region of the space of $(k, \gamma)$ where a colluding group could not be detected based on its size and edge density due to the existence of many similar groups of honest reviewers.

### 4.2 Detection Algorithm Evaluation (Q2)

For values of $(k, \gamma)$ larger than those that exist in the honest bidding, we may hope that colluding groups can be identified as suspicious by an appropriate algorithm. However, this may not always be possible: since the size of the colluding group is unknown, a detection algorithm must identify that the colluding group is more suspicious than the honest-reviewer groups of different sizes. In this section, we evaluate whether several existing techniques for dense-subgraph discovery suffice to detect colluding groups with larger values of $(k, \gamma)$, including one algorithm demonstrated to effectively detect fraud on Twitter.

We consider two desiderata in selecting algorithms to evaluate. First, the algorithms should not require explicitly specifying the size of the output subset. This is a requirement in our setting, since conference program chairs have no evidence regarding the size of the colluding groups that they can use to direct the detection algorithms. Second, our algorithms should be based on different notions of subgraph density that implicitly balance the size and edge density of the groups in a different way. Since $\mathcal{G}_1$ is unattributed, this choice is the primary design decision made by a density-based detection algorithm. By considering a variety of such choices, we hope to find one that detects the true colluders across the largest possible range of injected size and density. As a result, we consider the following algorithms:

- Traditional densest-subgraph discovery (Goldberg, 1984; Charikar, 2000): Output the subset of vertices $\mathcal{S}$ that maximizes $f(\mathcal{S}) = |\mathcal{E}_1[\mathcal{S}]|/|\mathcal{S}|$. This corresponds to the subgraph with highest average degree. In our case, since we count all edges in both directions, this is a generalization of the standard definition for undirected graphs. To solve this problem, we implement the LP-based exact algorithm from Charikar (2000). We refer to this as "DSD".

- Optimal quasi-clique discovery (Tsourakakis et al., 2013): For a given parameter $\alpha \in [0, 1]$, output the subset of vertices $\mathcal{S}$ that maximizes the "edge surplus" $f(\mathcal{S}) = |\mathcal{E}_1[\mathcal{S}]| - 2\alpha\binom{|\mathcal{S}|}{2}$. The second term in $f(\mathcal{S})$ corresponds to the expected number of edges in the subgraph if each edge occurs independently with probability $\alpha$. Since we count all edges in both directions, we include the factor

of 2 in the second term as compared to the standard definition for undirected graphs. To solve this problem, we implement the two approximation algorithms proposed by Tsourakakis et al. (2013), one based on greedy peeling and one based on local search. We refer to these algorithms as "OQC-Greedy" and "OQC-Local" respectively. As recommended by Tsourakakis et al. (2013), we set $\alpha = 1/3$.

- TellTail (Hooi et al., 2020): This method first defines the "adjusted mass" of a subset as the difference between the number of edges in the subset and the expected number of edges in the subset if edges are rewired randomly (preserving vertex degrees). Subsets are then scored based on the probability of the adjusted mass under a fitted Generalized Pareto distribution. Most parameters of this distribution are fixed as constants based on empirical observations across several datasets. Since this method operates on undirected graphs and is non-trivial to adapt to a directed setting, we first map our bidding graph $\mathcal{G}_1$ to an undirected graph $\mathcal{G} = (\mathcal{V}_1, \mathcal{E})$ before inputting it to this algorithm. The input graph $\mathcal{G}$ has the same vertex set as $\mathcal{G}_1$ and has an edge $(r_1, r_2)$ iff both edges $(r_1, r_2) \in \mathcal{E}_1$ and $(r_2, r_1) \in \mathcal{E}_1$; that is, $\mathcal{E} = \{(r_1, r_2) \in \mathcal{R}^2 : (r_1, r_2) \in \mathcal{E}_1 \wedge (r_2, r_1) \in \mathcal{E}_1\})$. Denote the degree of a vertex $v$ in $\mathcal{G}$ by $deg(v)$ and the CDF of the Generalized Pareto distribution as $F_{GP}$. Concretely, the algorithm finds a subset of vertices $\mathcal{S} \subseteq \mathcal{V}$ that approximately maximizes the objective function $f(\mathcal{S}) = F_{GP}\left(|\mathcal{E}[\mathcal{S}]| - \frac{\sum_{v \in \mathcal{S}} deg(v)}{4|\mathcal{E}|}\right)$. Note that this algorithm implicitly takes into account the connectivity between the chosen subset and the rest of the graph.

DSD, OQC-Greedy, and OQC-Local operate on pre-specified notions of density, without considering the sparsity of subgraphs in the input graph. In contrast, TellTail defines density in a data-driven fashion by identifying the distribution of masses that subgraphs of certain sizes follow, arguing that other notions of density tend to be biased towards larger subgraphs. TellTail was also shown to detect fraudulent followers on Twitter. These algorithms cover a representative set of the landscape of dense-subgraph mining solutions.

We evaluate the detection algorithms against the following collusion model. For each setting of $(k, \gamma)$, we choose a subset of $k$ reviewers uniformly at random from among those reviewers that authored at least one paper. We then add edges uniformly at random between colluding reviewers until the subgraph has edge density at least $\gamma$. This modified graph is then passed as input to each detection algorithm. We repeat this procedure for 50 trials for each setting. We report the mean Jaccard similarity between the set of injected colluders and the set of reviewers output by the detection algorithms.

For the detection algorithms that use local search to optimize their objective (OQC-Local and TellTail), we return the result with highest objective value over 11 initializations. The first run is initialized according to the triangle-counting heuristic suggested by Tsourakakis et al. (2013), and the remaining 10 runs are initialized uniformly at random. We find that the detection performance of OQC-Local is significantly better when only the heuristic initialization is used, since the random initializations often resulted in output with a higher objective value but lower overlap with the true colluders; thus, we show the results with heuristic initialization only in this section and defer the results with all initializations to Appendix B. However, the poor performance of OQC-Local when all initializations are used indicates that the objective value of OQC-Local is misaligned with the detection objective.

The results for DSD, OQC-Local, and TellTail are shown in Figure 2. Results for OQC-Greedy are shown in Appendix B as it generally performs worse than OQC-Local. In both datasets, DSD performs very poorly; this is because it consistently returns a overly large subset of reviewers. On the AAMAS dataset, OQC-Local does somewhat better, achieving Jaccard similarity above 0.5 for values of $k \geq 26$ and $\gamma \geq 0.9$. TellTail performs by far the best, detecting colluders consistently for moderate-to-large values of $(k, \gamma)$. However, there exists a wide range of settings in which all algorithms fail to detect injected colluders: for example, reviewer groups with $8 \leq k \leq 12$ and $\gamma = 1.0$ do not exist in the honest bidding and yet are still not detected by any algorithm. On the S2ORC dataset, OQC-Local and TellTail appear to perform equally well. There is a similar region in which honest-reviewer groups do not exist and yet detection fails to identify the correct group with high probability (e.g., $10 \leq k \leq 12$ with $\gamma = 1.0$), albeit smaller than in the AAMAS dataset.

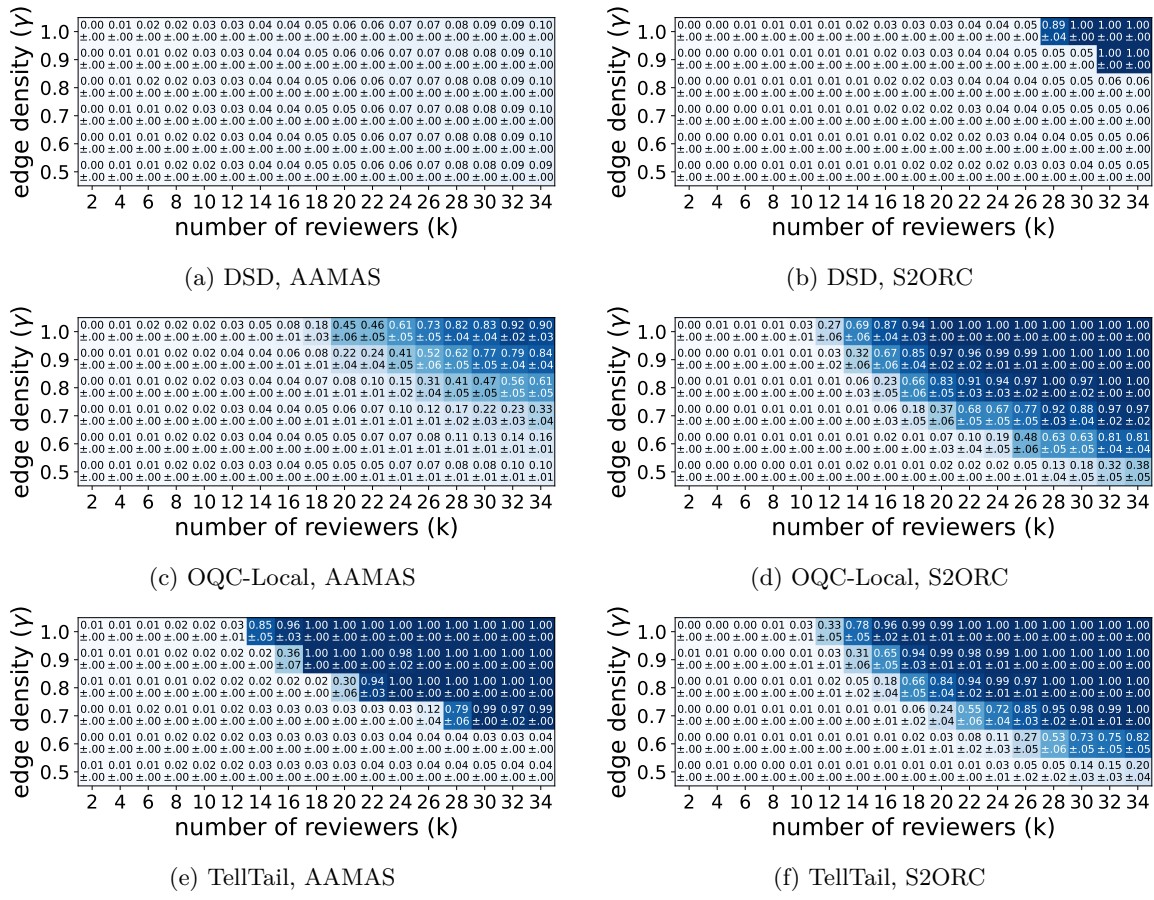

Figure 2: Performance of detection algorithms on $\mathcal{G}_1$. Values indicate the mean Jaccard similarity between the true set of colluders and the algorithm output, along with standard errors. Higher values correspond to better detection performance.

## 4.3 Manipulation Success Evaluation (Q3)

In this subsection, we evaluate the impact of collusion in terms of the colluders' success at achieving their desired paper assignments, contextualizing the results of the previous two subsections. We conduct experiments in which we inject colluding groups into $\mathcal{G}_1$ for each setting of the parameters $(k, \gamma)$ in the exact same manner as in Section 4.2. We then fix this modified version of $\mathcal{G}_1$ (i.e., the input to the detection algorithms in Section 4.2) as the "target graph" and modify $\mathcal{B}$ in order to realize this graph.

However, there are many possible strategies that these colluders could employ to modify their bids that correspond to the addition of these edges, since each edge $(r_1, r_2)$ denotes the presence of at least one bid from reviewer $r_1$ on a paper authored by reviewer $r_2$. Additionally, the relationship between bids and edges is not one-to-one due to co-authorship between reviewers, which means that exactly realizing the target graph via bidding may not be possible. Due to these challenges, we instead modify the bids of colluders to achieve a modified version of $\mathcal{G}_1$ that is "no more suspicious" than the target graph. Specifically, we consider each edge $(r_1, r_2)$ in the target graph such that $r_1$ is in the injected colluding group. If $r_2$ is also a colluder, we add bids from $r_1$ on each paper authored by $r_2$; otherwise, we choose one existing bid from $r_1$ on a paper authored by $r_2$ uniformly at random and remove all other bids from $r_1$ on papers authored by $r_2$. In this way, we ensure that the edges within the injected colluder group are a subset of the edges in the target graph and the edges outside the injected colluder group are a superset of the edges in the target graph. Subject to these constraints, this procedure allows colluders to bid according to a worst-case strategy.

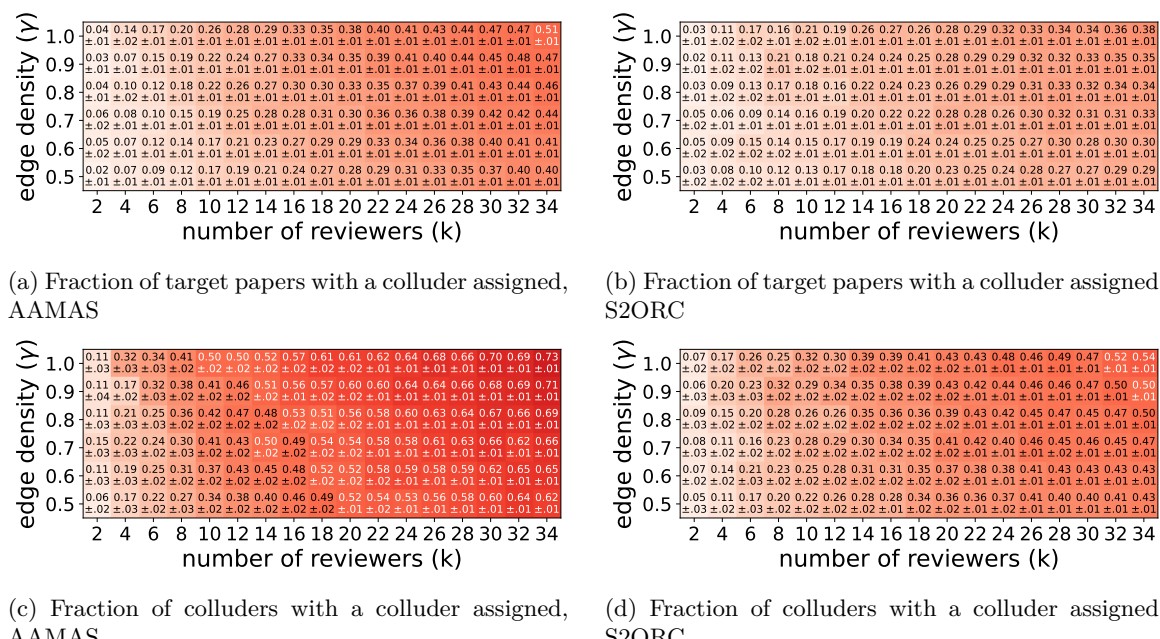

(a) Fraction of target papers with a colluder assigned, AAMAS

(b) Fraction of target papers with a colluder assigned, S2ORC

(c) Fraction of colluders with a colluder assigned, AAMAS

(d) Fraction of colluders with a colluder assigned, S2ORC

Figure 3: Success of colluders in terms of $k$ and $\gamma$. Values indicate the mean for each metric along with standard errors.

Given the modified bids, we compute a paper assignment as detailed in Section 3.1. Since the exact objective of colluders is not obvious, we evaluate the success of the malicious reviewers in two ways. First, we count the fraction of "target" papers (i.e., those authored by at least one colluder) with at least one colluder assigned; this indicates the extent to which colluders succeeded at influencing the acceptance decision for all of their papers. This may be too harsh of a metric, since colluders may not aim to influence the decision for all of their papers simultaneously. Second, we count the fraction of colluders who authored at least one paper that has at least one other colluder assigned; this indicates the fraction of colluders who recieved some benefit from participating in the collusion ring. We conduct 50 trials for each setting of the parameters and report the mean for each of the two metrics.

The results are shown in Figure 3. For the results on the AAMAS dataset, of particular interest are the results with $k \leq 20$ and $\gamma \leq 0.8$, since these were not detected by any of the algorithms in Section 4.2. We see that at the extremes of this region, approximately one-third of colluder-authored papers have at least one colluder assigned, and approximately one-half of colluders have at least one colluder assigned to one of their papers. At $(k = 10, \gamma = 0.8)$, where a larger number of honest-reviewer groups exist, both success metrics are also reasonably high. The results on the S2ORC dataset are similar: for example, at $(k = 14, \gamma = 0.8)$, 22% of target papers and 35% of colluders are successfully targeted while all detection algorithms achieve low success rates. Similarly, 21% of papers and 34% of colluders are successfully targeted when colluders are camouflaged among numerous honest-reviewer groups at $(k = 12, \gamma = 0.9)$. Thus, colluders can influence the acceptance decisions for a moderate fraction of their submitted papers without being detected by the evaluated algorithms. The counts of honest-reviewer groups in Section 4.1 suggest that a portion of the success of colluders may be unavoidable in this setting, regardless of the strength of the detection algorithms. In practice, the implication of these results is that any group of reviewers could decide to collude and unethically improve their chances of getting a paper accepted at the conference, tainting the fairness of the peer review process, while remaining completely safe from bidding-based detection.

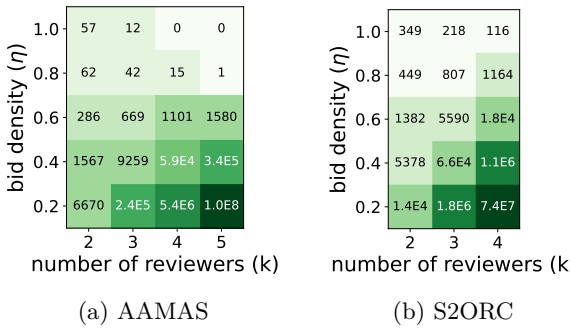

Figure 4: Exact counts of honest-reviewer groups with varying size and bid density.

# 5 Bipartite Bidding Graph

In this section, we represent the reviewer bidding data as a bipartite graph, in which one set of vertices corresponds to reviewers and the other set of vertices corresponds to papers. This graph contains three types of undirected edges between a reviewer $r$ and a paper $p$: bid edges, indicating that reviewer $r$ bid positively on paper $p$; authorship edges, indicating that reviewer $r$ authored paper $p$; and conflict-of-interest edges, indicating that reviewer $r$ has a conflict-of-interest with paper $p$ but did not author it. Our motivation for considering this graph is that it directly represents all data that the detection algorithms have access to.

Formally, we denote this graph as $\mathcal{G}_2 = (\mathcal{V}_2, (\mathcal{E}_2^{(B)}, \mathcal{E}_2^{(A)}, \mathcal{E}_2^{(C)}))$, where $\mathcal{V}_2 = \mathcal{R} \cup \mathcal{P}$ is the vertex set and $\mathcal{E}_2^{(B)} = \mathcal{B}$, $\mathcal{E}_2^{(A)} = \mathcal{A}$, and $\mathcal{E}_2^{(C)} = \mathcal{C} \setminus \mathcal{A}$ are the sets of bid, authorship, and conflict edges respectively. As in Section 4, we characterize a (potentially colluding) group of reviewers $\mathcal{S} \subseteq \mathcal{R}$ in terms of a size parameter $k_{\mathcal{S}} = |\mathcal{S}|$ and a density parameter $\eta_{\mathcal{S}}$. In $\mathcal{G}_2$, we consider a new notion of density, which we call "bid density". For any subset of reviewers $\mathcal{R}' \subseteq \mathcal{R}$, define $\mathcal{P}[\mathcal{R}'] = \cup_{r \in \mathcal{R}'}\{p \in \mathcal{P} : (r, p) \in \mathcal{A}\}$ to be the subset of papers authored by at least one reviewer in $\mathcal{R}'$. The bid density of $\mathcal{S}$ is then defined as the total number of bids made by reviewers in $\mathcal{S}$ on papers in $\mathcal{P}[\mathcal{S}]$, divided by the maximum possible number of bids by reviewers in $\mathcal{S}$ on papers in $\mathcal{P}[\mathcal{S}]$:

$$\eta_{\mathcal{S}} = \frac{|\mathcal{E}_2^{(B)}[\mathcal{S} \cup \mathcal{P}[\mathcal{S}]]|}{|\mathcal{S}||\mathcal{P}[\mathcal{S}]| - |\mathcal{E}_2^{(A)}[\mathcal{S} \cup \mathcal{P}[\mathcal{S}]]| - |\mathcal{E}_2^{(C)}[\mathcal{S} \cup \mathcal{P}[\mathcal{S}]]|}.$$

We again will omit the subscript $\mathcal{S}$ from $k_{\mathcal{S}}$ and $\eta_{\mathcal{S}}$ since it will be clear from context.

We now analyze the problem of detecting collusion from $\mathcal{G}_2$. As in Section 4, the following three subsections present empirical analysis aimed at answering the three research questions identified in Section 3.3.

## 5.1 Honest-Reviewer Groups (Q1)

First, we count the number of honest-reviewer groups in $\mathcal{G}_2$ for each value of size $k$ and bid density $\eta$. We consider only reviewers that have authored at least one paper. Since the bid density for a group is naturally lower than the edge density in general, we consider values of bid density ranging between 0.2 and 1.0.

The counts are shown in Figure 4. Due to the more complicated definition of bid density, we cannot efficiently prune branches of the search tree and must enumerate all $\binom{|\mathcal{R}|}{k}$ reviewer subsets. As a result, the range of feasible $k$ are significantly more limited than in Section 4.1: counts for $k \geq 6$ for AAMAS and $k \geq 5$ for S2ORC could not be completed in 24 hours. In S2ORC, we see that numerous honest-reviewer groups do exist at every feasible setting, up to $\eta = 1.0$ and $k = 4$. Honest-reviewer groups are less prevalent in AAMAS, but still exist at $k = 5$ and $\eta \leq 0.8$. Like the figures in Section 4.1, these figures may be independently useful to conferences as indicators of the number of false positives for any heuristic defenses against collusion they use.

As in Section 4.1, we also run a greedy peeling method to heuristically find high-density reviewer groups of larger sizes. However, this method performs very poorly and so we relegate the results to Appendix B.

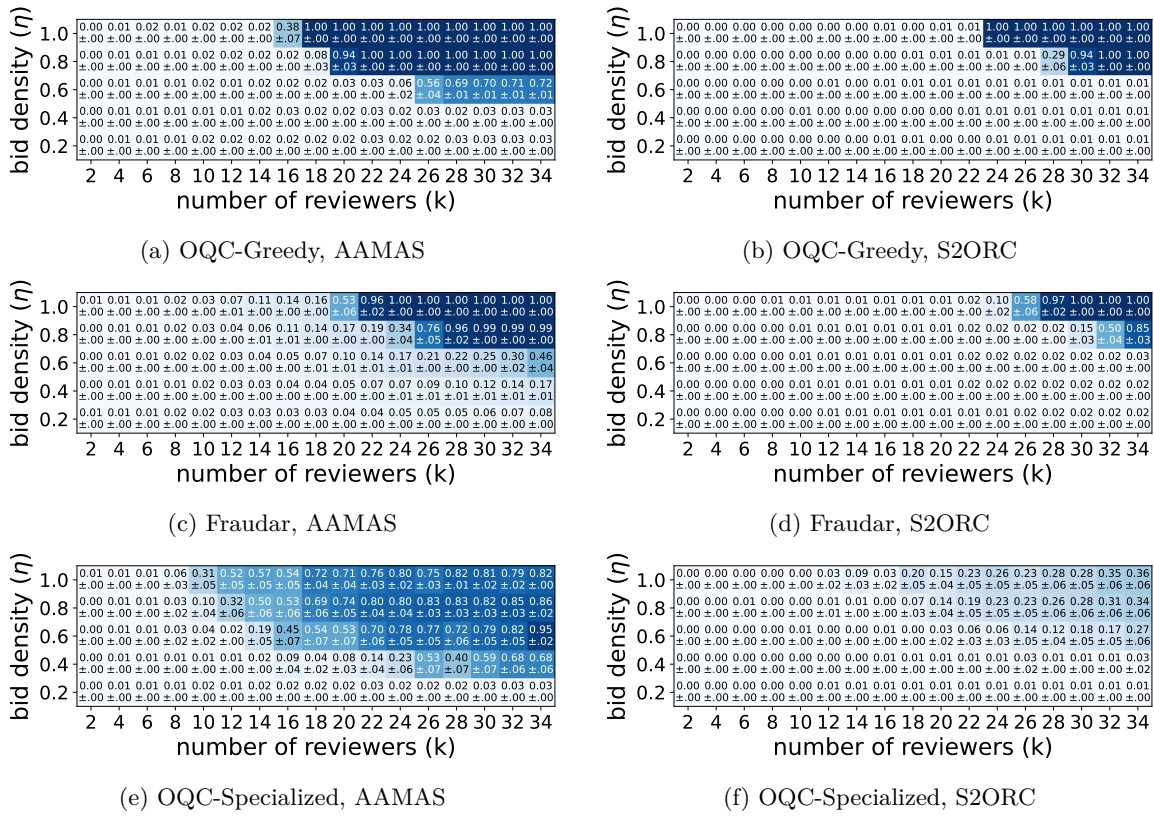

Figure 5: Performance of detection algorithms on $\mathcal{G}_2$. Values indicate the mean Jaccard similarity between the true set of colluders and the algorithm output, along with standard errors. Higher values correspond to better detection performance.

## 5.2 Detection Algorithm Evaluation (Q2)

Next, we evaluate the performance of a variety of dense-subgraph discovery algorithms at detecting injected collusion in $\mathcal{G}_2$, including an influential fraud-detection algorithm for the online review setting. Specifically, we consider the following algorithms:

- DSD, OQC-Greedy, OQC-Local, and TellTail: These algorithms are introduced in Section 4.2. As input to each algorithm, we provide an unattributed bipartite graph $\mathcal{G} = (\mathcal{V}_2, \mathcal{E})$ with the same vertex set as $\mathcal{G}_2$. We consider two variants for the edge set of the input graph: the edge set consists of only the bid edges $\mathcal{E} = \mathcal{E}_2^{(B)}$, and the edge set consists of the union of the bid and authorship edges $\mathcal{E} = \mathcal{E}_2^{(B)} \cup \mathcal{E}_2^{(A)}$. In the OQC algorithms, we use the original objective function for undirected graphs $f(\mathcal{S}) = |\mathcal{E}[\mathcal{S}]| - (1/3)\binom{|\mathcal{S}|}{2}$.

- Fraudar (Hooi et al., 2016): This algorithm is designed to detect fake product reviews (e.g., on Amazon/Yelp) or fake followers (e.g., on Twitter) from a bipartite graph of users and products $\mathcal{G} = (\mathcal{V}, \mathcal{E})$, where $\mathcal{E}$ contains edges from users to products. Denote by $deg(v)$ the degree of a vertex $v$ in $\mathcal{G}$. The returned subset of vertices $\mathcal{S} \subset \mathcal{V}$ is chosen to approximately maximize the objective function $f(\mathcal{S}) = \frac{1}{|\mathcal{S}|} \sum_{(u,v) \in \mathcal{E}[\mathcal{S}]} (\log(5 + deg(v)))^{-1}$. This objective is "camouflage-resistant", meaning that fraudulent users cannot affect their own objective value by adding extra edges to honest products. In our setting, we consider the reviewers to be users and papers to be products. We again input an unattributed bipartite graph with the same vertex set as $\mathcal{G}_2$ and two variants for the edge set: $\mathcal{E} = \mathcal{E}_2^{(B)}$ and $\mathcal{E} = \mathcal{E}_2^{(B)} \cup \mathcal{E}_2^{(A)}$.

- OQC-Specialized: We additionally adapt the objective function of the optimal quasi-clique discovery algorithm to our setting in a natural way. For a subset of reviewers $\mathcal{S} \subseteq \mathcal{R}$, we define the objective function

$$f(\mathcal{S}) = |\mathcal{E}_2^{(B)}[\mathcal{S} \cup \mathcal{P}[\mathcal{S}]]| - \alpha\Big(|\mathcal{S}||\mathcal{P}[\mathcal{S}]|$$
$$- |\mathcal{E}_2^{(A)}[\mathcal{S} \cup \mathcal{P}[\mathcal{S}]]| - |\mathcal{E}_2^{(C)}[\mathcal{S} \cup \mathcal{P}[\mathcal{S}]]|\Big).$$

Analogous to the concept of "edge surplus" that motivates the optimal quasi-clique objective function, this corresponds to the number of excess bids above the expectation if each bid is made independently with probability $\alpha$. We use local search to optimize this objective. As in the other OQC algorithms, we set $\alpha = 1/3$.

Since we aim to detect a group of colluding reviewers, we discard any paper vertices in the subset output by each algorithm, considering only the output reviewer vertices as the detected reviewers.

In addition to the algorithms from Section 4.2, Fraudar is an influential fraud-detection algorithm that leverages a density-based signal for detection. It was designed to detect fraudulent reviews on platforms like Amazon, a similar problem to our own. OQC-Specialized was not proposed in prior work, but naturally generalizes the objective function of OQC-Local to operate directly on $\mathcal{G}_2$.

We now describe the collusion model against which we evaluate the detection algorithms. For each value of $(k, \eta)$, we inject a group of colluding reviewers into $\mathcal{G}_2$ as follows. We first choose a set of $k$ reviewers, denoted by $\mathcal{M}$, uniformly at random from among all reviewers who have authored at least one paper. We then choose a reviewer-paper pair uniformly at random from $\mathcal{M} \times \mathcal{P}[\mathcal{M}]$ and add this pair to $\mathcal{E}_2^{(B)}$ if there is no existing edge of any type between this pair. We repeat this until the bid density is at least $\eta$. This modified graph is then used to construct the input to each detection algorithm. We repeat this procedure for 50 trials for each setting of parameters. We report the mean Jaccard similarity between the set of injected colluders and the output set of reviewers from the detection algorithms.

As in Section 4.2, for each local search algorithm (OQC-Local, TellTail, OQC-Specialized), we return the result with highest objective value over 11 initializations. The first initial subset is chosen according to a generalization of the heuristic suggested by Tsourakakis et al. (2013). For each vertex, we count the number of bid-author-bid-author cycles that this vertex participates in and divide by the total bid- and authorship-degree of the vertex; we then choose the initial subset to be the maximum vertex according to this metric along with all of its neighbors. The remaining 10 runs are initialized uniformly at random.

We show results for the OQC-Greedy, Fraudar, and OQC-Specialized algorithms in Figure 5. Results for the remaining algorithms, which generally performed worse, are in Appendix B. For all algorithms (other than OQC-Specialized), we find that the performance is very similar when the input edge set is $\mathcal{E}_2^{(B)}$ and when it is $\mathcal{E}_2^{(B)} \cup \mathcal{E}_2^{(A)}$; thus, all results shown are for the case with edge set $\mathcal{E}_2^{(B)}$. On AAMAS, we see that both OQC-Greedy and Fraudar are perform very well for high values of $(k, \eta)$ (e.g., $k \geq 20, \eta \geq 0.8$). However, both of these algorithms fail to detect colluders entirely at more moderate values of $k$ (e.g., $k = 14, \eta = 1.0$). OQC-Specialized achieves some success for a much wider range of parameters, but fails to achieve Jaccard similarities above 0.9 even for extreme values of $(k, \eta)$; this may indicate that the true colluding group is not a local optimum for this algorithm's objective function. On S2ORC, performance of all algorithms is significantly worse. OQC-Greedy and Fraudar still identify the colluding groups for extreme values of $(k, \eta)$, but OQC-Specialized fails to consistently identify the colluders for any parameter values. Overall, there exists a significant region of moderate parameter values at which no algorithm achieves good detection performance.

## 5.3 Manipulation Success Evaluation (Q3)

Finally, we evaluate the success of the colluders at manipulating the assignment as a function of the parameters $(k, \eta)$. For each setting of these parameters, we inject a group of colluders as described in the preceding section and compute a paper assignment using the procedure detailed in Section 3.1. As in Section 4.3, we

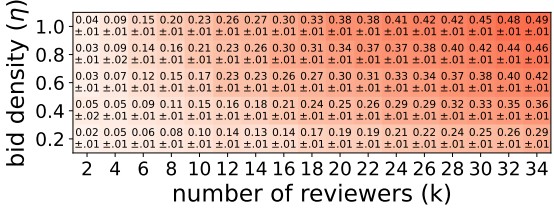

(a) Fraction of target papers with a colluder assigned, AAMAS

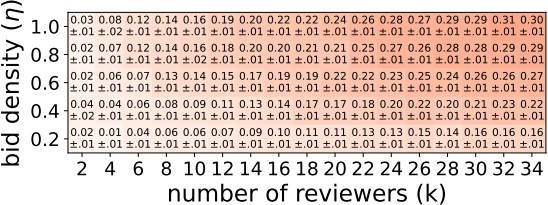

(b) Fraction of target papers with a colluder assigned, S2ORC

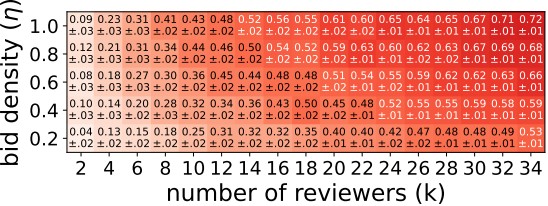

(c) Fraction of colluders with a colluder assigned, AAMAS

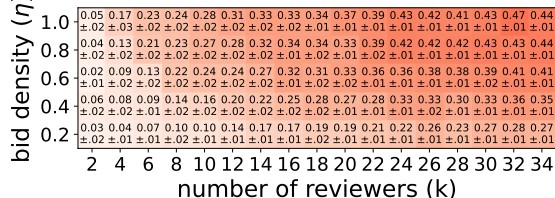

(d) Fraction of colluders with a colluder assigned, S2ORC

Figure 6: Success of colluders in terms of $k$ and $\eta$. Values indicate the mean for each metric along with standard errors.

evaluate the success of the colluders in two ways: the fraction of colluder-authored papers with at least one colluder assigned, and the fraction of colluders with at least one colluder assigned to one of their authored papers. We conduct 50 trials for each setting of the parameters and report the mean for each of the two metrics.

The results are shown in Figure 6. On AAMAS, we see that at $(k = 16, \eta = 0.8)$ where no detection algorithm was able to detect colluders with high probability, the colluders can successfully achieve assignments to 30% of their target papers and 54% of the colluders. Similar results are seen elsewhere on the frontier of the undetected region. On S2ORC, although the detection algorithms performed poorly, the success values are also lower than in the corresponding case on AAMAS. Still, in the cell $(k = 26, \eta = 0.8)$ where the detection algorithms performed poorly, colluders can achieve assignment to 26% of target papers and 42% of colluders, a sizeable influence on the paper assignment. We note that these success rates are quite similar to those found in Section 4.3 for the unipartite graph setting, which may provide some indication that the ability of undetected colluders to manipulate the assignment is robust to the exact graph representation. Overall, colluders are still able exert influence over the acceptance decisions for a non-trivial fraction of their own papers while avoiding detection.

## 6 Discussion

We provide an empirical exploration of the problem of detecting reviewer-author collusion rings from manipulated bidding, framing the problem as a dense-subgraph discovery problem in two different graph representations. Overall, we find that within our parametrized model of colluder behavior, malicious reviewers can manipulate the paper assignment to moderate success while remaining within typical or difficult-to-detect levels of density in the bidding graph (using popular existing anomaly-detection algorithms). This provides evidence to support the conclusion that malicious reviewer behavior cannot be effectively detected using just the bidding data. While our analysis cannot conclusively prove that bidding data is insufficient to detect collusion, our methodology thoroughly explores this problem from several different analyses of realistic conference data. Our work considers only a specific parametrized model of collusive behavior in which colluders receive reciprocal benefits within the conference in question. However, our results indicate that even this behavior cannot be detected. In reality, colluders may be more sophisticated in their attempts to avoid detection, making detection more challenging.

One limitation of our work is that our analysis is performed on semi-synthetic datasets, since real datasets containing bidding and authorship data are not publicly available. Thus, it is possible that the bidding in our datasets is not fully representative of real conference bidding: for example, in the AAMAS dataset, honest reviewers with abnormal bidding patterns may have opted-out of data collection. To further verify our results, program chairs could replicate our methodology on their own conference's real bidding data. Another limitation of our work is that we consider an intentionally narrow research question: the feasibility of detection from just the bidding, without the use of other metadata. As one of the first works on detecting collusion, our analysis of this question provides a basis for future work to build on.

While our results do not immediately direct any specific detection algorithm to be deployed in practice, caution should be taken to avoid negative societal impacts from the misuse of any detection algorithms deployed in the future. The cost of misidentifying an honest reviewer as part of a collusion ring can have enormous consequences for their reputation and career. To prevent such an instance, conference program chairs should at least conduct an associated thorough manual investigation before pursuing any form of action against a suspected collusion ring.

An important contribution of our work is to provide direction for future work. Most clearly, future work can consider the problem of detection in richer-featured settings than just the binary bidding setting we consider here and demonstrate if detection is feasible in such settings. Some examples of additional features that may be helpful are the strengths of bids, text-similarity scores, and past co-authorships. In these settings, future work can evaluate other types of anomaly detection methods beyond the dense-subgraph discovery algorithms we consider. This analysis would be very helpful for further directing the scope of future research on malicious bidding–if such research finds negative results even with additional metadata, research ought to continue to focus on mitigation-based techniques to address collusion rings. One major challenge for this line of research is the lack of availability of high-quality, fully-featured conference data. Future work can help address this need for datasets by collecting and releasing paper bidding data; one approach is to conduct an experiment from which (labeled) data can be collected, as done by Jecmen et al. (2023) for collusion and by Stelmakh et al. (2021) for a different problem of malicious behavior in peer review. Our work also provides some direction for research on collusion mitigation by establishing the region of colluding behavior that can be easily detected. Mitigation efforts need not be concerned with collusion in this region and can instead focus on mitigation within the undetectable region, particularly in the area where honest-reviewer groups do not exist.

### Acknowledgments

This work was supported in part by ONR N000142212181, NSF CAREER Award 1942124, NSF IIS 2200410, and NSF IIS 2310482.

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

## A    Text Similarity Details

In this section, we provide more details about the text-similarity scores in our datasets, introduced in Section 3.2. We synthetically generated the AAMAS text-similarity scores to supplement the original bidding dataset. The S2ORC text-similarity scores were included in the original dataset by Wu et al. (2021), and were computed by running the widely-used TPMS algorithm on the abstracts of the reviewer's past papers and the paper in question. Figure 7 shows the CDFs of the text-similarity scores among the reviewer-paper pairs that did not have conflicts-of-interest.

Since our work analyzes the effect of bids on the paper assignment, it is important to validate that our text similarity scores have a realistic level of agreement with the bids. We do this by comparing to the results of Stelmakh et al. (2023), who evaluate the accuracy of commonly-used text-similarity algorithms at predicting reviewer expertise using a ground-truth dataset. Specifically, Stelmakh et al. (2023) evaluate the accuracy of

the text-similarity scores by considering reviewer-paper-paper triples $(r, p_1, p_2) \in \mathcal{R} \times \mathcal{P} \times \mathcal{P}$ where reviewer $r$ reported greater expertise on paper $p_1$ than on paper $p_2$. They find that the TPMS algorithm produces a weakly greater text-similarity score for $(r, p_1)$ than for $(r, p_2)$ on 80% of "easy" triples (where the reviewer reported high-expertise vs low-expertise) and on 62% of "hard" triples (where the reviewer reported high-expertise vs higher-expertise). We use these results to generate and/or validate the text-similarity scores in our datasets, considering the bids to be a proxy for reviewer expertise.

For the AAMAS dataset, text-similarity scores were sampled i.i.d. from one of three Gaussian distributions, depending on the bid value for that reviewer-paper pair (from {"Yes", "Maybe", "No response"}). For the "No response" pairs, the Gaussian distribution was fit to the dataset of text-similarities reconstructed from the 2018 International Conference on Learning Representations by Xu et al. (2019). The variance of the Gaussian distributions for the "Maybe" and "Yes" pairs were the same, but the means were chosen based on the statistics from Stelmakh et al. (2023). Specifically, we set the means such that in expectation: (a) 80% of the triples $(r, p_1, p_2) \in \mathcal{R} \times \mathcal{P} \times \mathcal{P}$ where $r$ bid "Yes" or "Maybe" on $p_1$ and $r$ bid "No response" on $p_2$ had $T(r, p_1) \geq T(r, p_2)$; and (b) 62% of the triples $(r, p_1, p_2) \in \mathcal{R} \times \mathcal{P} \times \mathcal{P}$ where $r$ bid "Yes" on $p_1$ and $r$ bid "Maybe" on $p_2$ had $T(r, p_1) \geq T(r, p_2)$.

For the S2ORC dataset, we compare the text-similarity scores in the dataset to the statistics from Stelmakh et al. (2023). Recall that the S2ORC dataset contains bid values for each reviewer-paper pair in $\{0, 1, 2, 3\}$. We find that 83% of the triples $(r, p_1, p_2) \in \mathcal{R} \times \mathcal{P} \times \mathcal{P}$ where $r$ bid from $\{1, 2, 3\}$ on $p_1$ and $r$ bid 0 on $p_2$ had $T(r, p_1) \geq T(r, p_2)$. Additionally, we find that 65% of the triples $(r, p_1, p_2) \in \mathcal{R} \times \mathcal{P} \times \mathcal{P}$ where $r$ bid 3 on $p_1$ and $r$ bid $\{1, 2\}$ on $p_2$ had $T(r, p_1) \geq T(r, p_2)$. We see that the accuracy of the text-similarities in the S2ORC dataset is similar to that of the ground-truth dataset.

## B  Omitted Experimental Results

In Figure 8, we provide evaluations of detection algorithms on $\mathcal{G}_1$ omitted from Section 4.2. This includes the OQC-Greedy algorithm, as well as the OQC-Local algorithm with all initializations (detailed in Section 4.2). These algorithms generally perform worse than those with results shown in Section 4.2.

In Figure 9, we show the size and bid density of honest-reviewer groups detected by a greedy peeling method on $\mathcal{G}_2$ omitted from Section 5.1. In this method, we begin with the entire set of reviewers. At each iteration, we greedily remove the reviewer whose removal results in the highest bid density within the set of remaining reviewers. We then plot the bid density for each group size across the iterations. Clearly, this method performs very poorly at finding groups of high density.

In Figure 10, we present the results of the detection algorithms on $\mathcal{G}_2$ omitted from Section 5.2: DSD, OQC-Local, and TellTail. These algorithms generally perform worse than those with results shown in Section 5.2.

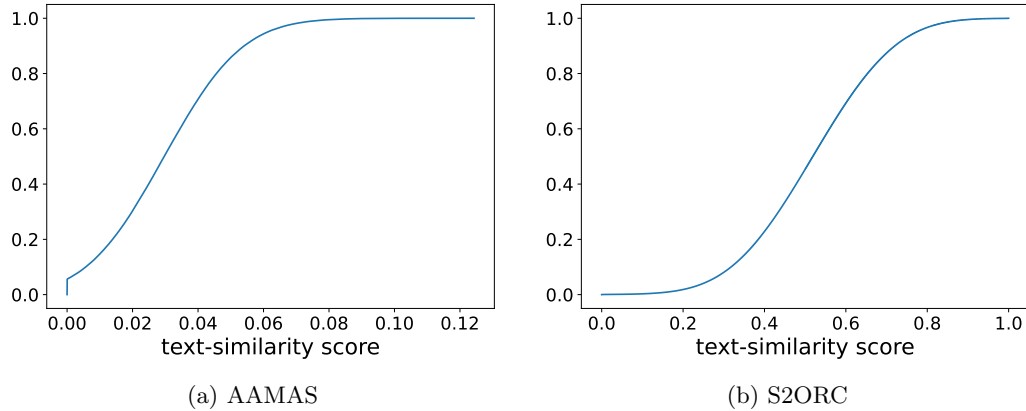

(a) AAMAS

(b) S2ORC

Figure 7: CDF of the text-similarity scores for reviewer-paper pairs without a conflict-of-interest. The mean text similarity score among these pairs is 0.030 for AAMAS and 0.52 for S2ORC.

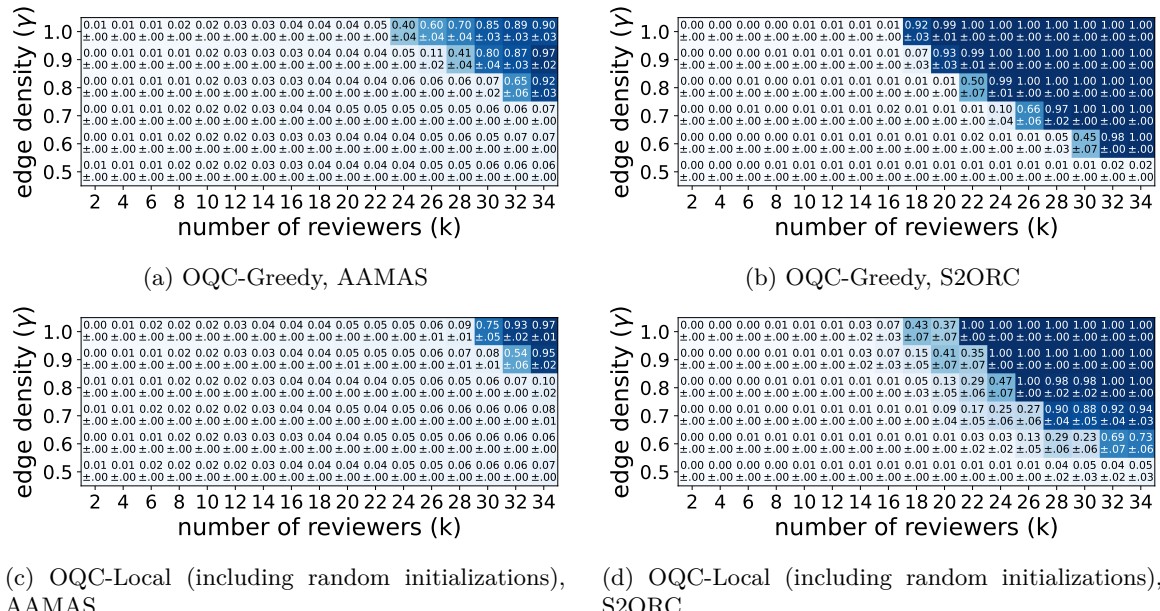

(a) OQC-Greedy, AAMAS

(b) OQC-Greedy, S2ORC

(c) OQC-Local (including random initializations), AAMAS

(d) OQC-Local (including random initializations), S2ORC

Figure 8: Performance of additional detection algorithms on $\mathcal{G}_1$. Values indicate the mean Jaccard similarity between the true set of colluders and the algorithm output, along with standard errors. Higher values correspond to better detection performance.

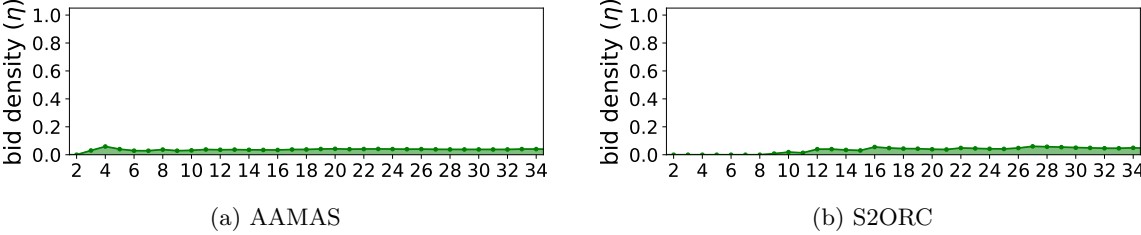

(a) AAMAS

(b) S2ORC

Figure 9: Size and bid density of honest-reviewer groups found by a heuristic method on $\mathcal{G}_2$. Each point corresponds to an existing honest-reviewer group (found by a greedy peeling method), and the shaded area indicates the region in which there exists at least one honest-reviewer group.

## C Density of Detected Groups

One question relevant to the analyses in Sections 4.2 and 5.2 is why the detection algorithms fail to detect the true colluders. Specifically, one may be interested in distinguishing the scenarios in which the detection algorithms output a group of reviewers less-dense than the true colluders (indicating that the colluders could potentially be detected if a better algorithm was developed) from the scenarios in which the detection algorithms output a group of reviewers with higher-density (indicating that it will be difficult for any algorithm to detect the colluders). In the following plots, we show the mean (edge or bid) density of the groups output by the detection algorithms when colluding groups of varying size and density are injected. Note that the detection algorithms are not restricted to outputting groups of the same size as the injected colluders.

The results for all algorithms are shown in Figure 11 (unipartite setting) and Figure 12 (bipartite setting). Overall, we generally see that in the (lower-left) region of the plots in which the algorithms fail to detect the true colluders (according to Figures 2 and 5), the algorithms are instead detecting reviewer groups with low density. This indicates that their objective may not be aligned well with the profile of the true colluders in this region. However, in the unipartite setting for the S2ORC dataset, TellTail detects a highly-dense group of honest reviewers when the colluding group is small and less-dense. This provides some additional

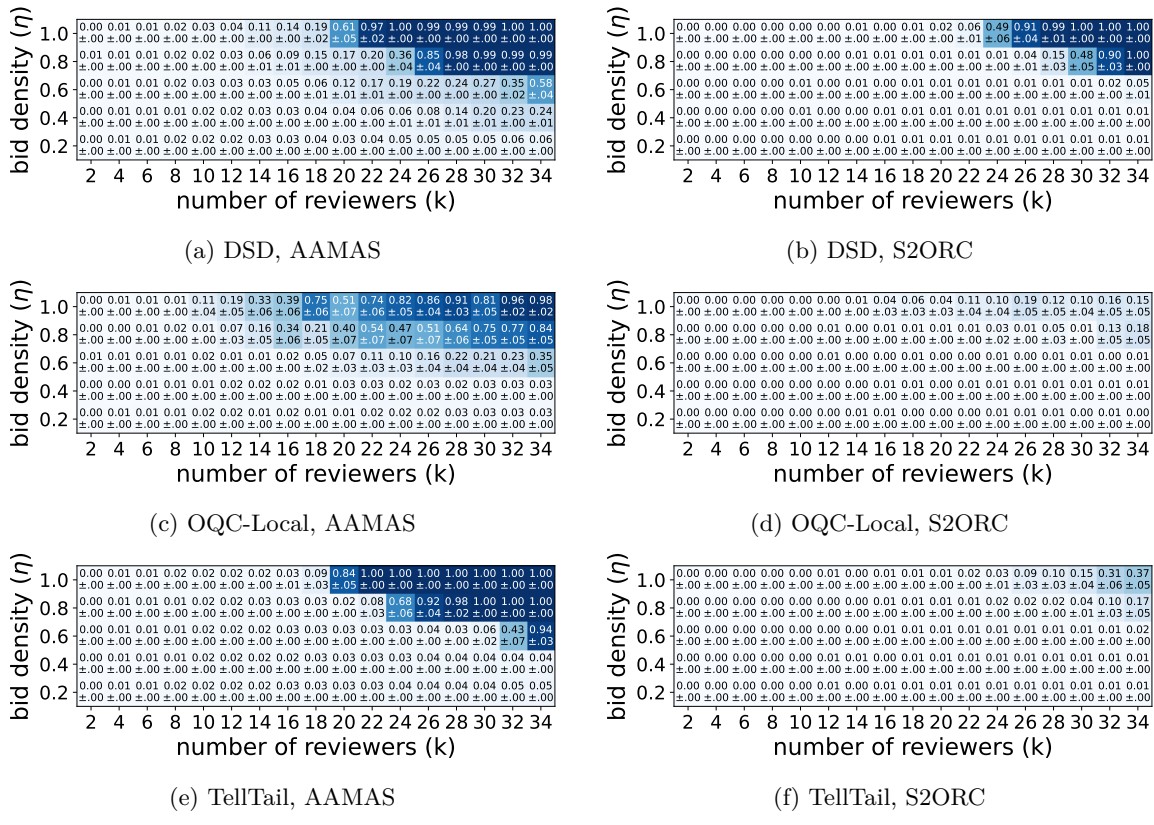

Figure 10: Performance of additional detection algorithms on $\mathcal{G}_2$. Values indicate the mean Jaccard similarity between the true set of colluders and the algorithm output, along with standard errors. Higher values correspond to better detection performance.

evidence that colluder detection in the region of ($k \leq 14, \gamma \leq 0.8$) is not feasible, due to the presence of a highly-dense group of honest reviewers that can mislead the detection algorithms.

## D  Experimental Results with Variant Datasets

As we detail in Section 3.2, the datasets we analyze in this work are semi-synthetic. The AAMAS dataset does not contain authorships, and so we construct authorships by subsampling three conflicts-of-interest for each paper. The S2ORC dataset contains bids that were synthetically constructed by Wu et al. (2021) based on bidding statistics from NeurIPS 2016. In this section, we vary these synthetic aspects and conduct additional experiments on these variant datasets.

In the variant AAMAS dataset, we construct authorships as follows. For each paper, instead of subsampling three conflicts-of-interest uniformly at random, we choose an integer from $\{1, \ldots, 5\}$ uniformly at random and subsample that many conflicts-of-interest to use as the authorships. In the variant S2ORC dataset, we modify the original bids by removing 15,000 positive bids (over 10% of the positive bids) uniformly at random and adding positive bids among 15,000 non-bidding pairs uniformly at random.

The results of the experiments on these variant datasets are shown in Figures 13-14 (unipartite setting) and Figures 15-16 (bipartite setting). In general, we see that the results are very similar to those obtained with the original datasets; for comparison, see Figures 2-3, Figures 5-6, and Figures 8-10. Notably, in the unipartite setting, the TellTail, OQC-Greedy, and OQC-Local detection algorithms perform slightly better on both variant datasets than on the original datasets. In the middle region where $k = 15$ and $\gamma \geq 0.8$, these algorithms are able to achieve moderate-to-high success rates, unlike on the original datasets.

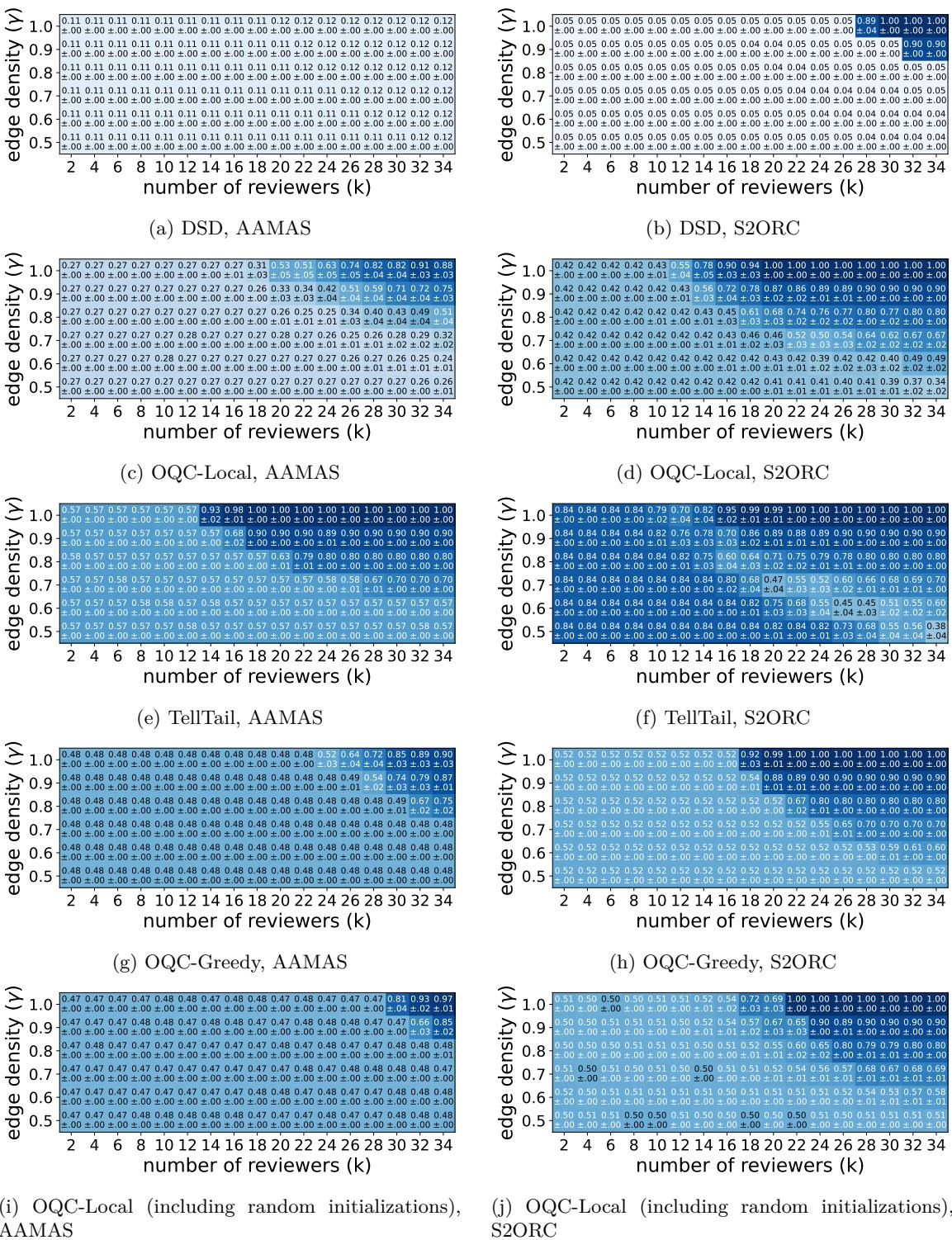

Figure 11: Density of detected groups on $\mathcal{G}_1$. Values indicate the mean edge density of the group of reviewers output by the algorithm, along with standard errors.

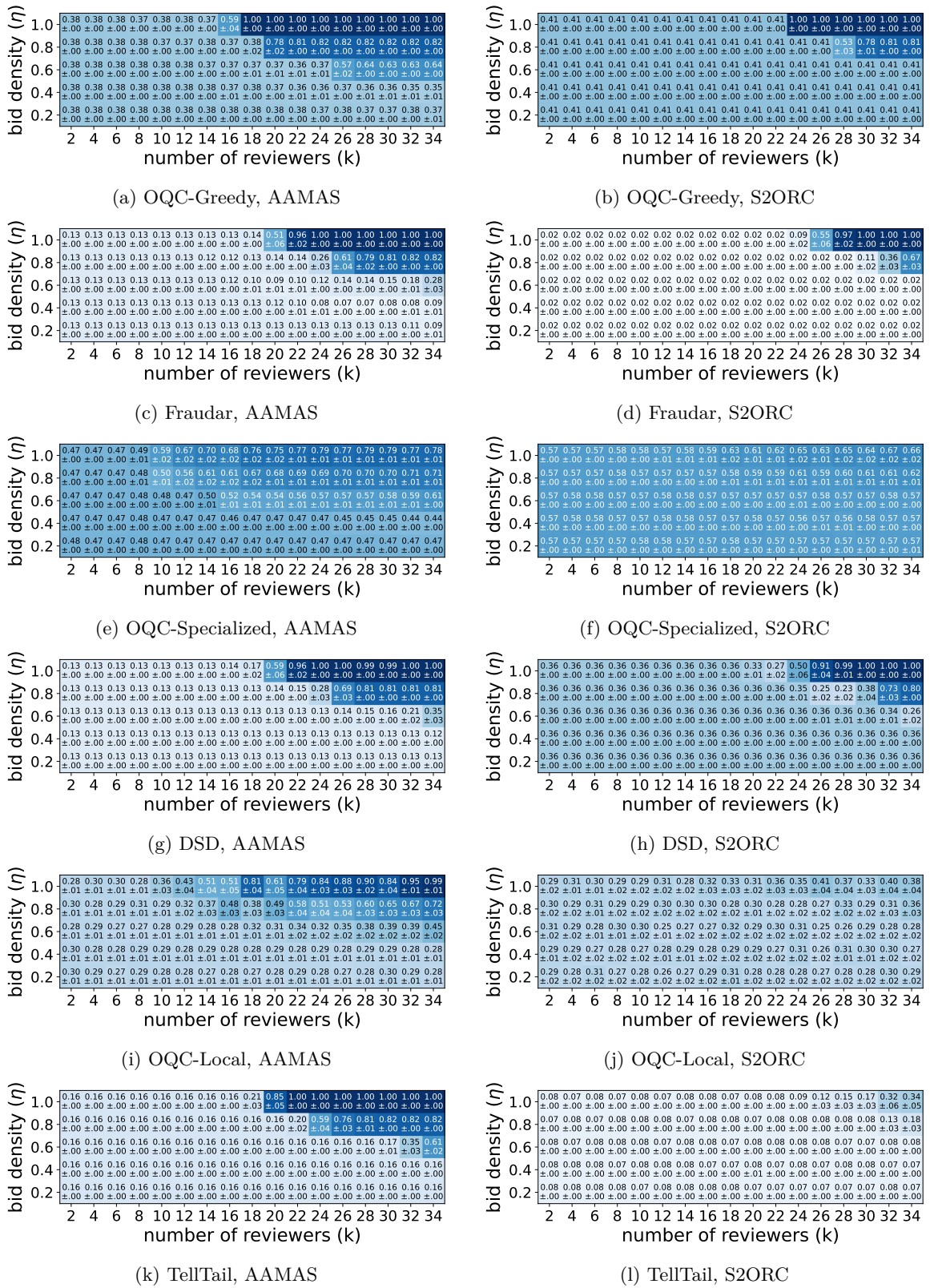

Figure 12: Density of detected groups on $\mathcal{G}_2$. Values indicate the mean bid density of the group of reviewers output by the algorithm, along with standard errors.

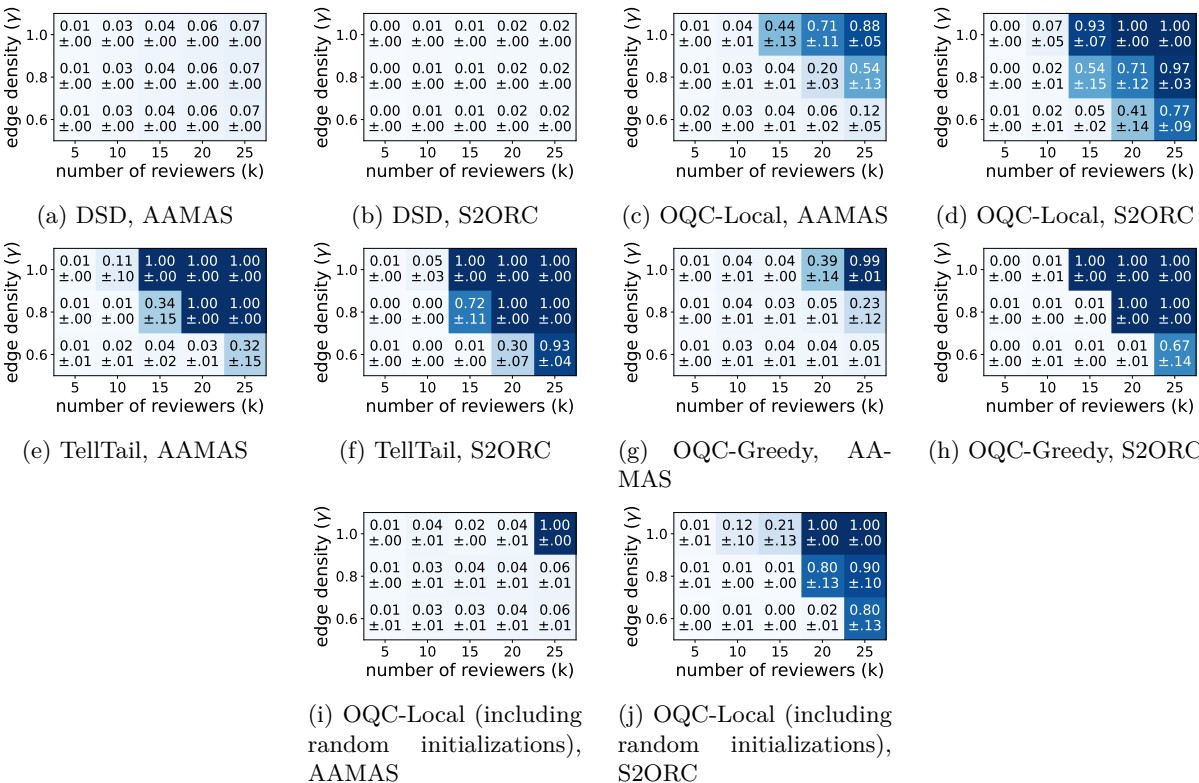

Figure 13: Performance of detection algorithms on $\mathcal{G}_1$ with variant datasets. Values indicate the mean Jaccard similarity between the true set of colluders and the algorithm output, along with standard errors.

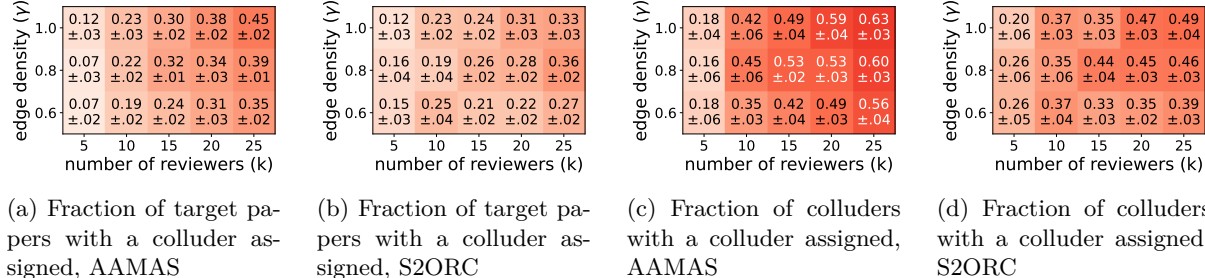

Figure 14: Success of colluders in terms of $k$ and $\gamma$ with variant datasets. Values indicate the mean for each metric along with standard errors.

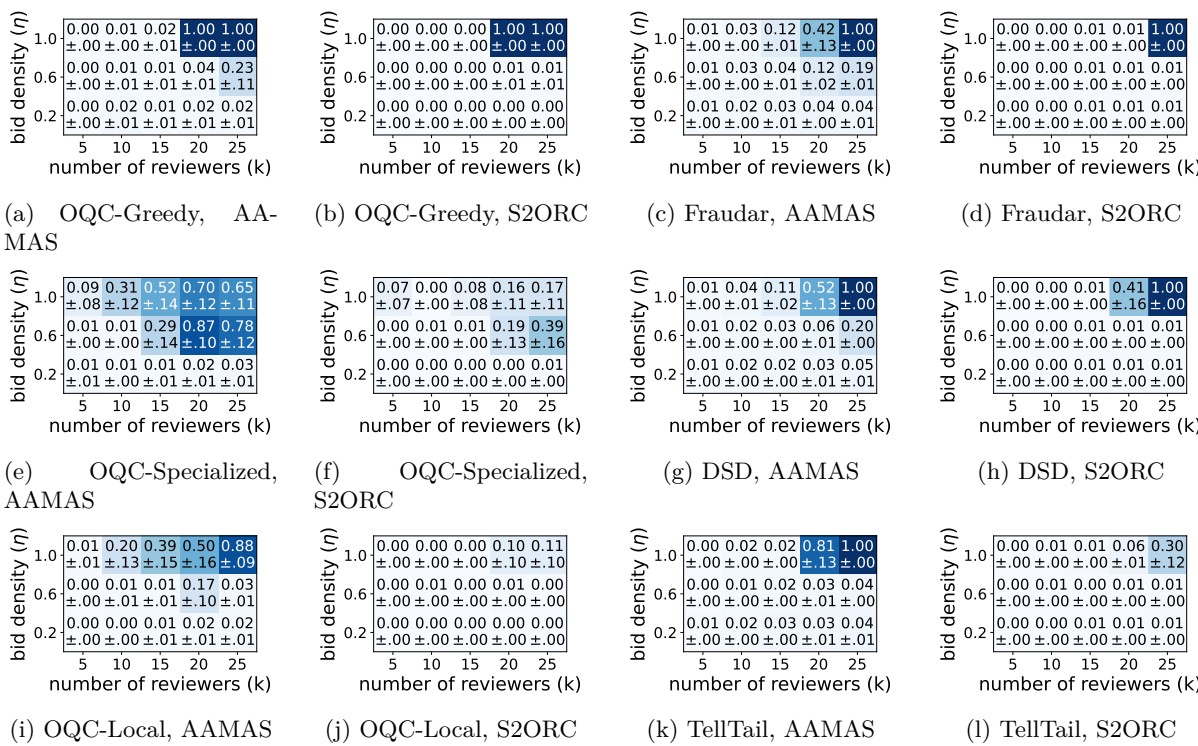

Figure 15: Performance of detection algorithms on $\mathcal{G}_2$ with variant datasets. Values indicate the mean Jaccard similarity between the true set of colluders and the algorithm output, along with standard errors.

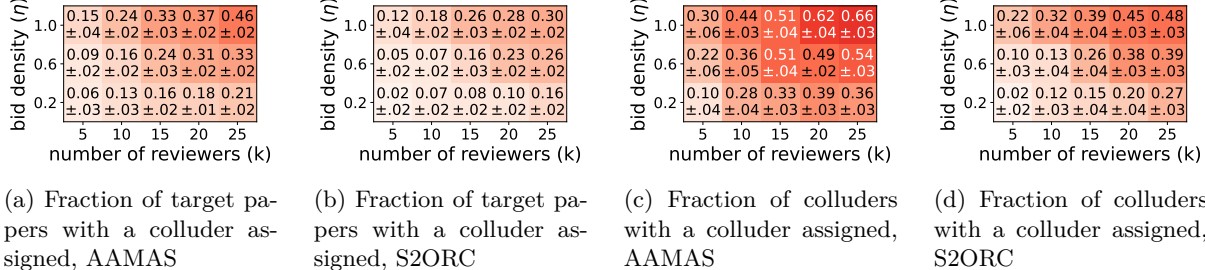

Figure 16: Success of colluders in terms of $k$ and $\eta$ with variant datasets. Values indicate the mean for each metric along with standard errors.

