# OpenReview forum: "On the Detection of Reviewer-Author Collusion Rings From Paper Bidding"
_TMLR — Accepted by TMLR_

### Review · Reviewer_9GQ8 · 2024-06-16

**Summary Of Contributions:**

This paper studies the problem of detecting collusive behavior in paper bidding at computer science conferences. The work attempts to detect simulated collusion in two datasets, using various off the shelf algorithms, and shows that collusive behavior cannot reliably be detected.

**Audience:**

Yes

**Broader Impact Concerns:**

No concerns.

**Claims And Evidence:**

Yes

**Requested Changes:**

Please see above.

**Strengths And Weaknesses:**

After receiving some pointers from the editor, I now recognize that I was not calibrated for TMLR when I wrote this review. In particular, I note in the review critieria: "We explicitly avoid [the] terms (“significant”, “impactful”, “novel”), and focus instead on the notion of “interest”." My review appealed to these forbidden concepts. My apologies.

In the end, it's of course up to the editor to accept or reject the paper, and it's fine with me if they reach a different conclusion than I did. I probably would not have taken the time to provide a peer review if I had realized at the outset that my professional judgement about the importance of the work wasn't being sought. I have no concerns about the work's correctness. It appears that what remains is a question about the extent to which the paper's claims are supported by the evidence.

The paper makes two broad kinds of claims. One kind is unproblematic: the following algorithm running on the following synthetic dataset yielded the following pattern of results. I have no concerns about any claims of this kind made in the paper. Of course, it's not altogether clear why anyone would care about such claims, given the substantial gaps between the questions that were studied and the phenomena motivating the paper. The paper certainly gives the impression that its findings have broader scope; see e.g. in the abstract:

"In this work, we tackle the question of whether it is feasible to detect collusion rings from the paper bidding. To answer this question, we conduct empirical analysis of two realistic conference bidding datasets, including evaluations of existing algorithms for fraud detection in other applications. We find that collusion rings can achieve considerable success at manipulating the paper assignment while remaining hidden from detection"

One might argue that this text doesn't constitute a "claim". The authors are fairly careful to offer caveats in their claims; nevertheless, one is left with the sense that the paper offers useful, actionable conclusions about the real world beyond having studied a limited, synthetic setting:

"These results suggest that collusion cannot be effectively detected from the bidding using popular existing tools"

"Overall, we find that malicious reviewers can manipulate the paper assignment to moderate success while remaining within typical or difficult-to-detect levels of density in the bidding graph. This provides evidence to support the conclusion that malicious reviewer behavior cannot be effectively detected using just the bidding data."

It seems to me that the concerns raised in my original review (if rephrased to avoid verboten words like "significance") speak to the question of whether these claims are justified: whether the experiments described in this paper support meaningful conclusions about the extent to which collusive behavior can be detected in real conferences. I agreed to review this paper because I was optimistic that I would learn something about the topic, but in the end was quite unconvinced that the study says much about real collusion, given (in part) its restriction to only bidding data, to certain algorithms, to substantially synthetic data, to a particular model of collusion behavior, to a simplified model of bidding, to collusion in which rewards are exchanged only within the system, and to a single scoring function.

Relatedly, the authors also make claims like:

"An important contribution of our work is to provide direction for future work."

It's not clear to me what this direction is supposed to be, beyond "it would be a good thing to relax the many limitations of our work". I don't see that as an important contribution so much as a drawback of the existing work.

Anyway, again apologies for misunderstanding the TMLR review criteria, and I hope this clarification is useful. I understand that the authors will already have seen my original review, so rather than editing it I've prepended these comments; my original review (which contains more technical detail) follows.

=========

Overall, I am not convinced that these conclusions are significant, and so I do not recommend publication at TMLR. The paper studies a given set of heuristic algorithms in a synthetic setting, under restricted assumptions about the data available to the algorithms and about how colluders will behave, and concludes that these methods are not effective at identifying collusive behavior. This is not a proof that realistic collusion cannot be detected in real data!

The paper makes a large number of highly restrictive assumptions that limit the importance of its analysis:
* Maybe the algorithms could be better
* Maybe the simulated data models the wrong behavior.
* Why restrict only to bids and authorships?
* Why restrict to one model of collusive behavior: it's a game theoretic setting, and behavior will surely change in response to detection strategies.
* Simulating only positive and negative bids is simplistic. I'm not convinced that multiple levels of review enthusiasm necessarily make the colluder's job easier; e.g., they may also make detection easier.
* It is not at all clear that colluders should all receive reciprocal benefit. Some reported examples of collusion involve senior people pressuring junior people to accept their papers. In such cases, the reciprocal benefits come outside the reviewing process (support for tenure; absence of retaliation; etc.)
* The paper studied one scoring function (many have been considered) and a simple matching algorithm based on min-cost flow that lacks any constraints that aim to mitigate collusion.

More fundamentally, the paper focuses on *detecting* collusion. This is a fundamentally hard objective; there is no way that (particularly absent other meta-data) collusive behavior should have a different signature from bidding from a clique of reviewers with similar interests. I am personally not convinced that detection is more important than mitigation. Surely the important goal here isn't moral policing, it's finding a way to run conferences that does not allow collusion rings to succeed.

Some other comments:

Do the authors investigate a trade-off between how easy the collusion is to detect and how effective it is in the first place? (Ineffectiveness comes both from probability someone else will be matched and probability your reviewing load will be assigned to something else.)

If the authors are willing to assume that colluding bidders would opt out of data collection, the opt-out field would be a great signal to use in detecting collusion.

An interesting question would be the extent to which various mitigation strategies proposed are effective at preventing the matching of colluders with papers, even in simulated data such as that studied here.

p7 claims the analysis would help conferences estimate the false-positive rates of their mitigation strategies. This is really implausible, and not only because in many cases the submission is not able even to compute these values, instead offering lower bounds. Nobody thinks that an enormous nummber of distinct reviewers are engaged in collusive behavior at conferences. So the actual number of potential matches to which a heuristic applies is a better estimate of the false positive rate than the numbers given here, not least because it would be based on a conference's own data rather than on simulated data or partial data from a long-ago AAMAS conference. Seems to me a conference should be comfortable that a mitigation strategy has a small risk of false positives if it applies only to a small number of potential matches overall, and otherwise should expect a large number of false positives.

The whole analysis of dense subgraphs seems to miss a fundamental issue. Some areas are popular, with many papers and reviewers, and others are not. Unpopular areas are likely to yield dense subgraphs organically. Calibrating the density of detected subgraphs by comparing to the popularity of the area might be useful.

Citations seemed heavily skewed towards those involving a certain set of coauthors. Various other relevant research is cited but its ideas are not engaged with as deeply.

---

> ### Author Response · Authors · 2024-08-18
> **Author Response to Reviewer 9GQ8 (Part 1/2)**
>
> We thank the reviewer for their comments and do appreciate the time they took to review the paper.
>
> $~$
>
> In response to some of the specific restrictions that the reviewer criticizes in their original review (and reiterates in their updated review):
>
> - *"restriction to certain algorithms"*: We are looking at some of the most popular algorithms in the literature and seeing if they can perform detection in this setting. We are unclear about the reviewer's expectations in this instance, and the review doesn't seem to provide specific guidance. It is practically impossible to try out all possible algorithms or all functions from bidding data to flagged reviewers/papers. We have clarified in the paper that our claims only involve the application of popular existing algorithms to our problem.
>
> - *“restriction to only bidding data”* / *“Why restrict only to bids and authorships?”*: First, it is very important to note that in practice many conferences – for instance, most conferences hosted on the popular HotCRP conference management system where, in fact, the first instances of collusion were caught – use only bids to determine the assignment (with the potential addition of some coarse subject-area matching). An additional reason for considering only bids in this work is to narrow the search space of the question we consider and minimize the number of assumptions we must make.
>
> - *“restriction to collusion in which rewards are exchanged only within the system”* / *“It is not at all clear that colluders should all receive reciprocal benefit. … In such cases, the reciprocal benefits come outside the reviewing process.”*: Yes indeed, and the collusions of the form you mention are *even harder* to detect since there is no reciprocating structure. Our results are already negative for the case when the collusion is within the same conference, and the additional hardness of outside-reciprocity further bolsters the results.
>
> - *"restriction to a particular model of collusion behavior"*:  The model of collusion that we analyze is the simplest form of colluding behavior, in which colluding reviewers make additional bids on each others’ papers uniformly at random. Our results indicate that even this behavior cannot be detected. In reality, colluders may be more sophisticated in their attempts to avoid detection, making detection more challenging.
>
> - *"restriction to a single scoring function"*: We are not sure what the reviewer means by “scoring function.”
>
> - *restriction on data*: We have conducted our analysis on the highest-quality datasets that we could find. The lack of both (i) data on true collusion and (ii) public, fully-featured conference bidding datasets is a fundamental challenge involved in analyzing the feasibility of collusion detection.
>
> $~$
>
> We next respond to other points from the reviewer’s original review.  Regarding the following comment:
> > This is a fundamentally hard objective; there is no way that (particularly absent other meta-data) collusive behavior should have a different signature from bidding from a clique of reviewers with similar interests. I am personally not convinced that detection is more important than mitigation. Surely the important goal here isn't moral policing, it's finding a way to run conferences that does not allow collusion rings to succeed.
>
> We certainly agree that detection is a fundamentally hard objective; the purpose of our work is to empirically assess the extent to which collusive behavior does have a different signature from honest bidding in such a way that can be effectively detected. For instance, it may be the case in practice that most groups of similarly-interested reviewers have a broader set of authors whose papers' they are interested in, while malicious cliques must bid in a more concentrated fashion in order to get assigned to their target papers. Our results generally indicate that this is not the case, as there do exist bidding strategies in which malicious reviewers can achieve their desired assignments without standing out to the detection algorithms.
>
> (response continued below)

---

> > ### Author Response · Authors · 2024-08-18
> > **Author Response to Reviewer 9GQ8 (Part 2/2)**
> >
> > (continued from above)
> >
> > We also don't argue that detection is inherently more important than mitigation. However, we do believe that research on this topic should not be forced to focus solely on mitigation, and that detection-based methods (that is, approaches based on explicitly identifying suspected colluders) have potential advantages over mitigation-based methods. First, in major conferences in practice a key way of mitigation is to modify the reviewer-paper assignment process: either by introducing randomization (Jecmen et al., 2020) or additional constraints (Leyton-Brown et al., 2022). While these make the process more robust to collusions, the downside is that they trade off some assignment quality for honest reviewers and papers. Now, instead suppose hypothetically that there was a high-accuracy detection algorithm. Then, the assignment could use it by disabling the reviewer-paper pairs picked out by the algorithm and leaving the rest untouched, thereby potentially achieving a much better tradeoff between assignment quality and robustness to collusion. Second, some conferences in practice have adopted the following procedure. After receiving the reviews, the conference identifies any suspicious reviews, then goes back to the bidding patterns to see if the bidding patterns pertaining to these suspicious reviewers/authors are statistical outliers. Any work on detection can also help in such a procedure, as detection algorithms are also implicitly hypothesis tests.
> >
> > $~$
> >
> > - *Regarding the tradeoff between how easy the collusion is to detect and how effective it is*: Yes, we discuss this question in Sections 4.3 and 5.3 by comparison to the results in Sections 4.2 and 5.2.
> >
> > - *Regarding whether our analysis can help conferences estimate the false-positive rates of their mitigation strategies*: We agree that if a heuristic applies to a large number of reviewer-groups, then it likely has many false positives since we believe collusion is rare. However, the remaining question is whether a heuristic that applies to only a few groups has false positives. For example, a standard anomaly detection approach is to assign an anomaly score to every group (e.g., the density of bids within the group) and rank the groups by this score. However, while simply declaring (for instance) the top 0.1% of these groups as anomalies would ensure that the heuristic applies to only a few groups, it does not show that the heuristic has a low false-positive rate. To make such a claim, one needs to estimate how likely these scores would be if there were no colluding groups, which cannot be observed purely within the dataset under question itself. In our case, while the distribution of bid density certainly varies by conference, our analysis provides a sense of what typical bidding densities look like for conferences to use as a reference point.
> >
> > - *Regarding the suggestion to calibrate density with respect to subject-area popularity*: We agree that algorithms that explicitly use the subject area of the reviewer and paper could be interesting to investigate, although our work focuses on algorithms that do not use such metadata. However, to the extent that the existence of subject areas is reflected in the bidding data as a community structure, some of the detection algorithms that we analyze should implicitly take this into account. Specifically, the anomaly scores of the TellTail and Fraudar algorithms we evaluate are affected not only by the density within a given subgraph but also by the connectivity of a subgraph to the rest of the graph, since these algorithms were designed to operate in settings that also have community structure (e.g., Twitter followers). TellTail defines the expected distribution of subgraph density as a function of the data, while Fraudar considers paper popularity by weighting edges inversely to the number of bids received by the paper.

---

> > > ### Comment · Reviewer_9GQ8 · 2024-11-13
> > >
> > > I'm not entirely convinced, but given the thoroughness of the authors' replies and the review criteria of TMLR, and after considerable thought, I'm weakly inclined towards acceptance. I do find pretty convincing the authors' argument that what I saw as restrictions of the usefulness of their approach, they see as a negative result about the effectiveness of detection. IMO it would strengthen the paper if this interpretation was further emphasized. I do note that this argument does remain weakened by the concerns I raised earlier: the authors show a "negative result" only in the case of particular algorithms, for a particular generating distribution of synthetic data, for a particular scoring function (way of quantify the benefit to the system of matching a particular reviewer with a particular paper, given the bids), etc.

---

### Review · Reviewer_iUJd · 2024-06-25

**Summary Of Contributions:**

The paper studies the problem of detecting collusion rings between reviewers in peer-reviewing systems based on reviewers' bids. The main approach here is to search for dense subgraphs of the bidding graph (i.e., a set of reviewers who bid on many of each other's papers). In particular, it analyzes the following three questions in two different models of the problem:

1.  How dense are the subgraphs in bidding datasets without collusion?
2.  Can existing fraud detection algorithms effectively identify collusion rings?
3.  How effectively can colluders manipulate the system?

**Audience:**

Yes

**Broader Impact Concerns:**

The paper already discusses the risk of collusion algorithms that output false positives at various places; not in a broader impact statement though.

**Claims And Evidence:**

Yes

**Requested Changes:**

See Weaknesses. In addition:

1. (the authors have satisfactorily clarified this issue in their response); In the experiments on detecting colluding reviewers, the best result over multiple initializations is returned. This is only possible if the ground truth is known, which will not be the case when "applying" the algorithm right? As a result, all the results discussed in these and other settings should be understood as best-case results? If this is correct, then this should be highlighted and discussed prominently and should be presented alongside average and worst-case findings.
2.  Why is text similarity not used for detecting collusion rings? What is the reason for explicitly disregarding this information source? Are there scenarios out there where bids are available but text similarity is not (the authors point to assignment algorithms that combine information from both sources to come up with an assignment)
3.  It would be nice to make the discussion of the differences between the two settings in terms of algorithms and results a bit more pronounced.
4.  It should be clarified how much the contribution of the paper goes beyond that of Jecmen et al. 2023 and in what exact ways the two papers do (and do not) overlap.

Questions that came to mind while reading that paper (no need to address them but could be used to guide extensions): How do the detection algorithms perform if there are multiple colluding sets of authors? Another possible way to extend the paper would be the customization of fraud detection algorithms to the collusion setting.

*Some minor comments:*

- Page 2: "Concretely, we consider the question: is it possible to effectively detect collusion rings from only the bids and authorships?" This might be a good point to briefly point to alternative approaches that are not pursued in the paper to clarify the scope of the paper and its relation to other approaches.

- Page 4: "(e.g.) Amazon."

- Page 6: "Formally, we denote the graph...": might be good to point out that the graph is directed to avoid confusion.

- Page 7 (y-axes): Are these values exact, rounded, or lower bounds?

- Page 9: "We then add edges uniformly at random between reviewers until the subgraph has edge density at least γ"- > This is referring to colluding reviewers, right?

- Page 10: "uniformly at random and remove all other such bids." What does "such" refer to?

- Page 11: What precisely are "target" papers (cf. line 5)

**Strengths And Weaknesses:**

**Strengths**:

1.  Dealing with collusion is a relevant problem for the ML community.
2.  The paper studies a new variant of the problem (detecting collusion) that has not received sufficient in the past.
3.  The paper is sound and the claims are almost all very clearly supported by experimental evidence (see point 1 in changes as the main issue regarding this criterion).
4.  The paper is very well-written, nicely structured, and a pleasant read.
    1.  I especially enjoyed that the empirical research questions are clearly identified, which gives the experimental analysis a clear goal and structure. The structure is supported by a good split of content between the appendix and the main body.
5.  The paper generates some insight that could be of general interest to the broader CS community (e.g., regarding the effectiveness of collusion)

**Weaknesses:**

The paper does not make any significant methodological contribution. It is a purely experimental effort that applies existing methods. At the same time, the experiments are of very limited technical and implementation difficulty. As such, the main contributions of the paper relevant to others are the take-aways from the experiments. While there are some generally interesting observations, in my personal opinion, the generated insights are not sufficient to warrant an independent publication in a top journal. In particular, I have two main problems with the experimental analysis (details below):

1.  The analysis is generally quite superficial. Trends/results are merely observed and not explained.
2.  Due to the used datasets, the generalizability of the experimental takeaways (the main contribution of the work) remains largely unclear.

*Style of Experimental Analysis*

The analysis of the experiments could be expanded in terms of the description of the results, the identification of trends, and the search for explanations of these trends (For instance, the results from the different sections could be linked better together. E.g., in the analysis from 4.2, how much possible room for improvement is there in light of the results from Section 4.1?). Generally, the paper only rarely asks "Why?" questions. There are numerous possibilities to deepen the analysis. Just to give one example, at the end of Section 4.3, when concluding that "approximately one-third of colluder-authored papers have at least one colluder assigned, and approximately one-half of colluders have at least one colluder assigned to one of their papers." it would be interesting to ask what the properties of these papers are and why only "so few" colluders benefit. I am not saying that these are questions that need to be addressed in the paper but some more in-depth analysis would be appreciated and would increase the relevance of the work.

*Generalizability*

Another issue is that the data used in the experiments is only semi-realistic. I am aware that there are no better datasets out there to be used; however, given that the experiments are the main focus of the paper it is worrying that it is unclear how much the results generalize beyond the specific settings examined in the paper. This concern gets strengthened because the data generation methods make numerous model assumptions (that might all be plausible but come without a justification and analysis of their impact). One way to address this issue could be to vary the generation methods slightly and see how this impacts the results. If results turn out to be robust, it would become more plausible that the results generalize.

---

> ### Author Response · Authors · 2024-08-18
> **Author Response to Reviewer iUJd (Part 1/2)**
>
> We thank the reviewer for their comments and suggestions. We have addressed each of the reviewer’s requested changes in our revised manuscript.
>
> - *Regarding the the style of experimental analysis*: We recall that our original submission discussed some “why” questions in Sections 4.2 and 5.2 to explain the poor performance of the DSD, OQC-Local (with random initializations), and OQC-Specialized algorithms. Based on the reviewer’s remarks, we conducted a new analysis of why the detection algorithms fail to detect the true colluders. Specifically, one may be interested in distinguishing the scenarios in which the detection algorithms output a group of reviewers less-dense than the true colluders (indicating that the colluders could potentially be detected if a better algorithm was developed) from the scenarios in which the detection algorithms output a group of reviewers with higher-density (indicating that it will be difficult for any algorithm to detect the colluders). **In Appendix C of the revised manuscript, we have added plots which show the mean (edge or bid) density of the groups output by the detection algorithms when colluding groups of varying size and density are injected.** Overall, we generally see that in the region of the plots in which the algorithms fail to detect the true colluders, the algorithms are instead detecting reviewer groups with low density, indicating that their objective may not be aligned well with the profile of the true colluders in this region.
>
> - *Regarding the generalizability*:
>   - We would like to note that while some model assumptions were required in the generation of our datasets (due to the lack of available fully-featured datasets), we do provide justification for these assumptions based on real data. For example, the synthetic text-similarity scores we construct for the AAMAS dataset are generated based on the text-similarities of the ICLR 2018 papers reconstructed by Xu et al. (2019). We also show that the text-similarity scores for both datasets have a similar level of agreement with the bids as in the reviewer preference data collected by Stelmakh et al. (2023); see Appendix A for details. The bids for the S2ORC dataset, generated in prior work by Wu et al. (2021), are designed in part based on distribution of bids in data from NeurIPS 2016. In addition, we will publically release the code for all of our experiments (currently available in the supplemental material) upon publication. We encourage conference program chairs with appropriate access to real bidding data to reproduce our analysis and provide additional evidence regarding the generalizability of our results.
>   - In addition, we have conducted new analyses in which we vary the synthetic aspects of the AAMAS (which has synthetic authorships) and S2ORC (which has synthetic bids) datasets. In the variant AAMAS dataset, we construct authorships as follows. For each paper, instead of subsampling three conflicts-of-interest uniformly at random, we choose an integer from {1, …, 5} uniformly at random and subsample that many conflicts-of-interest to use as the authorships. In the variant S2ORC dataset, we modify the original bids by removing 15,000 positive bids (over 10% of the positive bids) uniformly at random and adding positive bids among 15,000 non-bidding pairs uniformly at random. **We have included these results in Appendix D of the revised manuscript.** Overall, we find that the results are qualitatively very similar to those obtained with the original datasets, with the exception of a few algorithms in the unipartite setting that achieve slightly better performance on the variant datasets.
>
> - *Regarding the question of multiple instantiations*: For the algorithms that attempt to maximize an objective through local search, we perform multiple trials of the local search with different initializations and report the result with the best objective value (not the best detection performance). This can be done when applying the algorithm in practice. **We have revised the paper to clarify this point.**
>
> - *Regarding the use of text-similarity*: In practice, many conferences–for instance, most conferences hosted on the popular HotCRP conference management system (where, in fact, the first instances of collusion were caught)–use only bids to determine the assignment and text-similarity are not available in those settings. An additional reason for considering only bids in this work is to narrow the scope of the question we consider and minimize the number of assumptions we must make. Our approach of considering only bids allows us to more thoroughly sweep the space of malicious reviewer strategies without making significant assumptions on their behavior. We believe that addressing the question of detections with text-similarities is a very important direction for future work, but not one that we choose to address in our work (one of the first works on detecting collusion).
>
> (response continued below)

---

> > ### Author Response · Authors · 2024-08-18
> > **Author Response to Reviewer iUJd (Part 2/2)**
> >
> > (continued from above)
> >
> > - *Regarding the difference between the two settings*: The two settings we examine in Sections 4 and 5 are two different ways of representing a conference's bidding in graph form. As it's not clear a priori which setting will be more conducive to detection nor which setting's notion of density more accurately reflects the behavior of malicious reviewers (see discussion in Sections 4.0 and 5.0), we conduct our analysis in both settings. We apply a similar set of detection algorithms in both settings (applying an algorithm in both settings when possible) and observe generally similar results.
> >
> > - *Regarding the differences from Jecmen et al. (2023)*: The primary contribution of Jecmen et al. (2023) is to release a dataset containing malicious reviewer bidding data, collected from a mock conference bidding process. The authors provide analysis of this dataset, including categorizing the strategies used by malicious reviewers and evaluating the performance of three simple detection algorithms. This analysis is focused primarily on comparing the performance of the observed strategies at successfully manipulating the assignment and at avoiding detection by various heuristics. While our work performs similar analyses of malicious reviewer success and detection, our focus is on addressing the question of whether detection is feasible rather than on the properties of any particular malicious reviewer strategies. To this end, we consider a wider variety of established detection algorithms from the anomaly detection literature and consider a wide range of possible behavior for the malicious reviewers.
> >
> > - *Regarding the minor comments*: **We have revised the paper to incorporate these suggestions.**
> >   - Page 2: We have added the following to the paper. *“Our work does not attempt to investigate various other questions of interest to the problem of collusion detection: for example, comparing the effectiveness of detection- and mitigation-based approaches, conducting a game-theoretical analysis of the detection problem, or gathering new data on the behavior of real colluders.”*
> >   - Page 7: In Figures 1(a)-1(b), the y-axes are lower-bounds on the edge density of the detected groups (i.e., the values represent "the number of groups of size k with edge density at least $\gamma$"). In Figures 1(c)-1(d), the y-axis represents the exact edge density of the identified groups.
> >   - Page 9: Correct, edges are added between the k chosen colluding reviewers.
> >   - Page 10: "Such" refers to all other existing bids from $r_1$ on any paper authored by $r_2$.
> >   - Page 11: The set of "target" papers is the set of all papers with at least one colluder as an author.

---

> > ### Comment · Reviewer_iUJd · 2024-09-08
> >
> > Dear authors, thank you for your response and for revising the paper according to my suggestions. The response has clarified the motivation behind only using bidding data and how multiple instantiations were used. I also appreciate the additional experiments conducted in response to my two main points of criticism.

---

### Review · Reviewer_SFrt · 2024-07-25

**Summary Of Contributions:**

The paper studies the reviewer-author collusion detection from paper bidding. Specifically, the paper focuses on detecting the collusion rings. The threat model is that there is a group of reviewers, which are authors and bid each other's paper. Intuitively, it turns out the edge density among this group should be large and therefore the paper studies several group detections by optimizing the edge density and its variants. The empirical study shows that none of detections are effective. The attack can escape from the detections while attain a high success rate of getting the target the papers.

**Audience:**

Yes

**Claims And Evidence:**

Yes

**Requested Changes:**

Evaluation with any detection that can output multiple groups will be a solution for detecting the collusion group. However, the paper missed this kind of detection, which weakens the conclusion "All detection algorithms fail to detect some injected collusion rings".

**Strengths And Weaknesses:**

**Strengths**
1. The paper studies pure bid-based methods for detecting collusion groups and demonstrates that pure bid-based detection is ineffective. The aspect of the attack and corresponding defense is novel.
2. The empirical evaluation is solid: it evaluates the attack and detection on two (semi-)real datasets and studies the different sizes and densities of the collusion group.

**Weaknesses**

The detection only outputs one group and the evaluation is to calculate the similarity between the group and the ground-truth collusion group. This understates the power of detection. A stronger version of detection could output multiple groups and the detection can be successful if one of them guess the collusion group is mostly correct. This will introduce a trade-off between detection and assignment quality: more detection groups will increase the chance of successfully detecting the collusion group while influencing the assignment quality. The results of this trade-off should be inexplicitly related to the exact counts in Figure 1.

---

> ### Author Response · Authors · 2024-08-18
> **Author Response to Reviewer SFrt**
>
> We thank the reviewer for their comments and suggestions.
>
> *Regarding detection algorithms that output multiple groups*: While such algorithms would be more likely to output the colluding group, one concern with this type of algorithm is the significant cost of false positives in our setting. Conferences should rightly be hesitant to accuse a reviewer of collusion without a high degree of certainty, since false accusations may cause significant damage to a reviewer's reputation (see the discussion in Section 1). However, detection algorithms with a high false-positive rate could be applied as part of a collusion-mitigation approach: flag a large number of reviewer-paper pairs as potentially collusive, and avoid assigning these pairs in the assignment algorithm. In this case, the important question to answer is whether this approach is more effective than the other collusion-mitigation approaches that have been proposed in the literature (e.g., Jecmen et al., 2020; Wu et al., 2021).
>
> While we agree this is an interesting question, our work considers only detection algorithms that attempt to identify the colluding group as the "most suspicious" group of reviewers.

---

### Decision · Action_Editor_M47H · 2024-11-28

**Recommendation:** Accept with minor revision

**Comment:**

Overall, the reviewer appreciated your detailed response and paper updates. I am happy to recommend acceptance and ask for one minor update (see below).

Two reviewers mention that the evaluation and design of collusion detection algorithms used in this paper might be restrictive in that detection is "binary." The authors provide a reasonable justification regarding the practical aspects of this (and the impracticability of false positives). This leads the authors to conclude that "All detection algorithms fail." Unless I missed it, I suggest the paper should at least briefly discuss this issue. I found your response to reviewer SFrt useful, and something at the level would be sufficient.

I also suggest the authors consider the final comment from reviewer 9GQ8, but I will let them decide whether it needs to be addressed in the final version of their work.

**Audience:**

The reviewers agree that the paper's findings are interesting and can serve as a "good introduction for the community."

**Claims And Evidence:**

After the discussion and the authors' reply, the reviewers agree that the claims are mostly supported by (accurate, convincing, and clear) evidence. The reviewers suggest that the conclusion that all "collusion algorithms fail" is a bit strong, and they suggest contextualizing it.